# On the Hidden Waves of Image

## Abstract

In this paper, we introduce an intriguing phenomenon – the successful reconstruction of images using a set of one-way wave equations with hidden and learnable speeds. Each individual image corresponds to a solution with a unique initial condition, which can be computed from the original image using a visual encoder (e.g., a convolutional neural network). Furthermore, the solution for each image exhibits two noteworthy mathematical properties: (a) it can be decomposed into a collection of special solutions of the same one-way wave equations that are first-order autoregressive, with shared coefficient matrices for autoregression, and (b) the product of these coefficient matrices forms a diagonal matrix with the speeds of the wave equations as its diagonal elements. We term this phenomenon *hidden waves*, as it reveals that, although the speeds of the set of wave equations and autoregressive coefficient matrices are latent, they are both learnable and shared across images. This represents a mathematical invariance across images, providing a new mathematical perspective to understand images.

## 1 Introduction

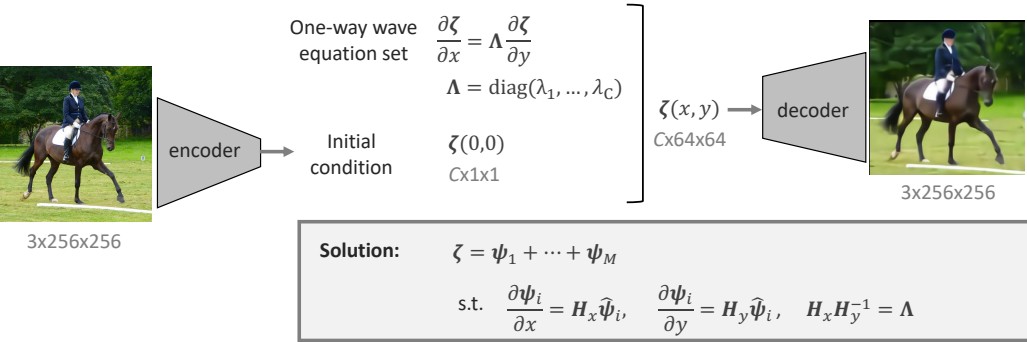

Figure 1: **Hidden waves phenomenon.** Each $256 \times 256$ image corresponds (to a good approximation) to a unique solution of one-way wave equations (or transportation equations) with an initial condition derived from the original image. The solution, with resolutions of $64 \times 64$ or $128 \times 128$, facilitates image reconstruction using a simple decoder consisting of upsampling and convolutional layers. The wave speeds, $\lambda_1, \lambda_2, \ldots, \lambda_C$, are latent and learnable. The solution $\boldsymbol{\zeta}$ is expressed as a sum of multiple special solutions, $\boldsymbol{\psi}_i$, which yield first-order autoregressive properties $\frac{\partial \boldsymbol{\psi}_i}{\partial x} = \boldsymbol{H}_x \hat{\boldsymbol{\psi}}_i$, $\frac{\partial \boldsymbol{\psi}_i}{\partial y} = \boldsymbol{H}_y \hat{\boldsymbol{\psi}}_i$ ($\hat{\boldsymbol{\psi}}_i$ is a normalized $\boldsymbol{\psi}_i$). The product of the coefficient matrices is a diagonal matrix with wave speeds, $\boldsymbol{H}_x \boldsymbol{H}_y^{-1} = \mathrm{diag}(\lambda_1, \lambda_2, \ldots, \lambda_C)$.

In recent years, the field of deep learning has emerged as a dominant force in the realm of computer vision, where it has taken the lead in numerous complex visual tasks, including image recognition He et al. (2016); Dosovitskiy et al. (2021); Touvron et al. (2020); Howard et al. (2017); Ma et al. (2018), object detection Ren et al. (2015); Lin et al. (2017); Carion et al. (2020), segmentation He et al. (2017); Chen et al. (2018a); Kirillov et al. (2023), and image generation Chen et al. (2020); Rombach et al. (2021); Ramesh et al. (2021), etc. Despite these impressive achievements, a fundamental question remains largely unexplored: *"What are the underlying mathematical properties shared by images?"* This question is of paramount importance as it delves into the heart of our understanding

of visual data and its representation. Unraveling the mathematical essence of images can potentially unlock deeper insights into their nature.

In this paper, we introduce an intriguing mathematical property of images, which we term the "*hidden wave phenomenon*" (illustrated in Figure 1). Specifically, images are closely approximated by a set of hidden one-way wave equations (or transportation equations) in the latent space. The term "*hidden*" refers to the fact that the speeds of waves ($\lambda_1, \ldots, \lambda_C$) are latent but learnable. Each image corresponds to a special solution with a unique initial condition that can be derived from the image through a convolutional neural network (CNN). Additionally, each image can be reconstructed from the corresponding special solution using a lightweight decoder composed of upsampling and $3\times 3$ convolutional layers.

Furthermore, we demonstrate an elegant method to achieve the solution of each image $\boldsymbol{\zeta}$. Firstly, it can be decomposed into multiple special solutions $\boldsymbol{\psi}_i$ as $\boldsymbol{\zeta} = \sum \boldsymbol{\psi}_i$. Secondly, each special solution $\boldsymbol{\psi}_i$ is first-order autoregressive, with partial derivatives (along $x$ and $y$ axes) dependent only on the current value $\boldsymbol{\psi}_i(x, y)$, as follows:

$$\frac{\partial \boldsymbol{\psi}_i}{\partial x} = \boldsymbol{H}_x \hat{\boldsymbol{\psi}}_i, \quad \frac{\partial \boldsymbol{\psi}_i}{\partial y} = \boldsymbol{H}_y \hat{\boldsymbol{\psi}}_i, \quad \boldsymbol{H}_x \boldsymbol{H}_y^{-1} = \boldsymbol{\Lambda} = \text{diag}(\lambda_1, \ldots, \lambda_C), \tag{1}$$

where $\hat{\boldsymbol{\psi}}_i$ is a normalized $\boldsymbol{\psi}_i$, and $\boldsymbol{H}_x$ and $\boldsymbol{H}_y$ are two $C \times C$ learnable matrices. Notably, $\boldsymbol{H}_x \boldsymbol{H}_y^{-1}$ forms a diagonal matrix with wave speeds $\lambda_1, \ldots, \lambda_C$ along the diagonal.

Our inspiration stems from recent research known as FINOLA Chen et al. (2023). This work has illuminated a remarkable insight: it demonstrates that all images can be represented within the feature space using a first-order norm+linear autoregressive model, enabling the successful recovery of the original image. Building upon this foundational idea, we introduce two significant extensions. Firstly, we reveal that through diagonalization, this representation transforms into a unique solution within a set of one-way wave equations (or transportation equations), subject to specific local constraints. Secondly, by replacing a single instance of FINOLA with a sum of multiple FINOLAs that share parameters, we relax the local constraints, resulting in significant improvements in image reconstruction. Notably, after relaxing the local constraints, the partial differential equations (PDEs) in FINOLA no longer hold. One-way wave equations become the new governing equations. Collectively, these two extensions give rise to the intriguing phenomenon we term *hidden waves*, as depicted in Figure 1.

While considering the overall framework, it is worth emphasizing that the encoder, decoder, and hidden waves are learned in an end-to-end manner. Our mathematical model underwent rigorous testing on ImageNet Deng et al. (2009), utilizing images of size $256\times 256$. By employing $C = 2048$ wave equations, we achieved a remarkable PSNR of 28.0 for image reconstruction on the validation set after just 100 training epochs. Moreover, we discovered that by allowing spatial shifts for the initial conditions of the special solution $\boldsymbol{\psi}_i$, we could attain the same outstanding PSNR performance while reducing the number of wave equations by half, with $C = 1024$.

In framing our research objectives, it is crucial to clarify that our pursuit does not revolve around achieving state-of-the-art performance. Instead, our primary aim is to shed light on a fundamental mathematical property shared by all images. This property manifests as a set of wave equations in a latent space, with hidden speeds and distinctive initial conditions. We aspire to foster a deeper understanding of images within the research community.

## 2 HIDDEN WAVES OF IMAGES

In this section, we will delve into the intricacies of the *hidden waves* phenomenon. Inspired by FINOLA Chen et al. (2023), we introduce two significant extensions: (a) the generalization of FINOLA to a set of one-way wave equations and (b) the relaxation of local constraints, resulting in more accurate reconstruction.

### 2.1 REVIEW OF FINOLA

Let's begin by reviewing the recent work by Chen et al. (2023) (referred to as FINOLA) that demonstrates the successful reconstruction of an image (size $256\times 256$) from a single vector $\boldsymbol{q}$ through two

steps: (a) placing $\boldsymbol{q}$ at the center to generate a feature map $\boldsymbol{z}(x, y)$ via first-order norm+linear autoregression, and (b) employing a simple decoder comprising upsampling and convolutional layers. Mathematically, the first-order norm+linear autoregression is represented as:

$$
\begin{aligned}
\boldsymbol{z}(x+1, y) &= \boldsymbol{z}(x, y) + \boldsymbol{A}\boldsymbol{z}_n(x, y) \\
\boldsymbol{z}(x, y+1) &= \boldsymbol{z}(x, y) + \boldsymbol{B}\boldsymbol{z}_n(x, y)
\end{aligned}
\quad where \quad \boldsymbol{z}_n(x, y) = \frac{\boldsymbol{z}(x, y) - \mu_{\boldsymbol{z}}}{\sigma_{\boldsymbol{z}}} \quad, \tag{2}
$$

where the matrices $\boldsymbol{A}$ and $\boldsymbol{B}$ are learnable and possess dimensions $C \times C$. They are shared across all image positions $(x, y)$ and images. The normalization of the feature map $\boldsymbol{z}$ involves subtracting the mean $\mu_{\boldsymbol{z}}$ and dividing by the standard deviation $\sigma_{\boldsymbol{z}}$ of the $C$ channels at each position $(x, y)$. If we substitute $x+1$ and $y+1$ with $x+\Delta x$ and $y+\Delta y$, where $\Delta x = \Delta y = 1$, Equation 2 transforms into a difference equation. Further extension involves considering infinitesimal values for $\Delta x$ and $\Delta y$, leading to the formulation of partial differential equations (PDEs):

$$
\begin{aligned}
\frac{\boldsymbol{z}(x+\Delta x, y) - \boldsymbol{z}(x, y)}{\Delta x} &= \boldsymbol{A}\boldsymbol{z}_n(x, y) \\
\frac{\boldsymbol{z}(x, y+\Delta y) - \boldsymbol{z}(x, y)}{\Delta y} &= \boldsymbol{B}\boldsymbol{z}_n(x, y)
\end{aligned}
\quad \xrightarrow{\Delta x \to 0, \, \Delta y \to 0} \quad
\begin{aligned}
\frac{\partial \boldsymbol{z}}{\partial x} &= \boldsymbol{A}\boldsymbol{z}_n \\
\frac{\partial \boldsymbol{z}}{\partial y} &= \boldsymbol{B}\boldsymbol{z}_n
\end{aligned}. \tag{3}
$$

Please take into account that the introduction of partial differential equations (PDEs) represents a theoretical extension of FINOLA from a discrete grid to continuous coordinates. However, establishing theoretical proof in this context is challenging. Empirical evidence supporting the effectiveness of FINOLA across diverse grid sizes, from $16 \times 16$ to $128 \times 128$, can be found in Chen et al. (2023).

## 2.2 GENERALIZATION OF FINOLA TO ONE-WAY WAVE EQUATIONS

We extend FINOLA by introducing a generalization that encompasses a set of one-way wave equations (or transportation equations), subject to two specific conditions: (a) the matrix $\boldsymbol{B}$ is invertible, and (b) the matrix $\boldsymbol{A}\boldsymbol{B}^{-1}$ is diagonalizable. Importantly, these two conditions have been empirically validated on ImageNet, and we provide further elaboration on these details below.

**Diagonalization:** Firstly, we can rewrite Eq. 3 as follows:

$$
\frac{\partial \boldsymbol{z}}{\partial x} = \boldsymbol{A}\boldsymbol{B}^{-1}\frac{\partial \boldsymbol{z}}{\partial y} = \boldsymbol{V}\boldsymbol{\Lambda}\boldsymbol{V}^{-1}\frac{\partial \boldsymbol{z}}{\partial y}, \tag{4}
$$

where the matrix $\boldsymbol{A}\boldsymbol{B}^{-1}$ is diagonalized as $\boldsymbol{V}\boldsymbol{\Lambda}\boldsymbol{V}^{-1}$. The column vectors of $\boldsymbol{V}$ constitute a basis of eigenvectors. The diagonal entries of $\boldsymbol{\Lambda}$ represent the corresponding eigenvalues, i.e., $\boldsymbol{\Lambda} = \text{diag}(\lambda_1, \lambda_2, \ldots, \lambda_C)$.

**One-way wave equations:** Next, we project the feature map $\boldsymbol{z}$ using the inverse of the eigen-matrix $\boldsymbol{V}^{-1}$, denoted as $\boldsymbol{\zeta} = \boldsymbol{V}^{-1}\boldsymbol{z}$. This transformation simplifies Eq. 4 to the following form:

$$
\frac{\partial \boldsymbol{\zeta}}{\partial x} = \boldsymbol{\Lambda}\frac{\partial \boldsymbol{\zeta}}{\partial y}, \quad \frac{\partial \zeta_k}{\partial x} = \lambda_k \frac{\partial \zeta_k}{\partial y}, \tag{5}
$$

where $\zeta_k$ represents the $k^{th}$ element of vector $\boldsymbol{\zeta}$, and $\lambda_k$ is the $k^{th}$ eigenvalue in $\boldsymbol{\Lambda}$. After projecting $\boldsymbol{z}$ onto $\boldsymbol{\zeta}$, each dimension $\zeta_k$ follows a *one-way wave equation* (also known as a transportation equation), where the rate of change along the $x$-axis is $\lambda_k$ times the rate of change along the $y$-axis. Its solution takes the form $\mathcal{F}_k(\lambda_k x + y)$, where $\mathcal{F}_k(\cdot)$ can be any differentiable function. Typically, one-way wave equation involves time $t$, here we replace it with $y$.

**FINOLA as a special solution:** When we combine the equations from Eq. 2 to Eq. 5, we find that the linear projection of the feature map $\boldsymbol{z}$ in FINOLA, denoted as $\boldsymbol{\zeta} = \boldsymbol{V}^{-1}\boldsymbol{z}$, represents a special solution of one-way wave equations. This special solution comes with an initial condition, $\boldsymbol{\zeta}(0, 0) = \boldsymbol{V}^{-1}\boldsymbol{q}$, and a distinctive local constraint outlined as follows:

$$
\begin{aligned}
\text{solve} \quad & \frac{\partial \boldsymbol{\zeta}}{\partial x} = \boldsymbol{\Lambda}\frac{\partial \boldsymbol{\zeta}}{\partial y}, \quad \boldsymbol{\Lambda} = \text{diag}(\lambda_1, \lambda_2, \ldots, \lambda_C) \\
\text{initial condition:} \quad & \boldsymbol{\zeta}(0, 0) = \boldsymbol{V}^{-1}\boldsymbol{q} \\
\text{local constraint:} \quad & \frac{\partial \boldsymbol{\zeta}}{\partial x} = \boldsymbol{H}_x\hat{\boldsymbol{\zeta}}, \quad \frac{\partial \boldsymbol{\zeta}}{\partial y} = \boldsymbol{H}_y\hat{\boldsymbol{\zeta}}, \quad \boldsymbol{H}_x\boldsymbol{H}_y^{-1} = \boldsymbol{\Lambda},
\end{aligned} \tag{6}
$$

where the matrices $\boldsymbol{H}_x$ and $\boldsymbol{H}_y$, and the normalization $\hat{\boldsymbol{\zeta}}$ are computed as follows:

$$\boldsymbol{H}_x = \boldsymbol{V}^{-1}\boldsymbol{A}, \quad \boldsymbol{H}_y = \boldsymbol{V}^{-1}\boldsymbol{B}, \quad \hat{\boldsymbol{\zeta}} = \frac{(C\boldsymbol{I} - \boldsymbol{J})\boldsymbol{V}\boldsymbol{\zeta}}{\sqrt{\boldsymbol{\zeta}^T(C\boldsymbol{I} - \boldsymbol{V}^{-1}\boldsymbol{J}\boldsymbol{V})\boldsymbol{\zeta}}}, \tag{7}$$

where $C$ represents the number of channels $\boldsymbol{\zeta}(x, y) \in \mathbb{C}^C$, $\boldsymbol{I}$ represents the identity matrix, and $\boldsymbol{J}$ represents an all-ones matrix.

**Local constraint and autoregression:** The local constraint specified in Eq. 6 exhibits a notable similarity to the one in Eq. 3, indicating a correlation between the first-order derivatives $\partial\boldsymbol{\zeta}/\partial x$, $\partial\boldsymbol{\zeta}/\partial y$, and the current value $\boldsymbol{\zeta}$. This local constraint plays a crucial role by allowing us to compute the entire solution $\boldsymbol{\zeta}(x, y)$ from the initial condition at a single position, $\boldsymbol{\zeta}(0, 0)$, using a straightforward first-order autoregression approach.

Traditionally, in one-way wave equations, the initial condition is specified along the $x$ axis, typically as $\boldsymbol{\zeta}(x, y = 0)$. However, in our context, the local constraint simplifies the initial condition specifically at the origin.

**Correlation between coefficients and wave speeds:** Notably, there's an elegant correlation between the coefficient matrices $\boldsymbol{H}_x$ and $\boldsymbol{H}_y$ and the wave speeds $\boldsymbol{\Lambda}$. This correlation is expressed as $\boldsymbol{H}_x\boldsymbol{H}_y^{-1} = \boldsymbol{V}^{-1}\boldsymbol{A}\boldsymbol{B}^{-1}\boldsymbol{V} = \boldsymbol{\Lambda}$.

**Is FINOLA the optimal solution?** Despite its advantageous properties, such as first-order autoregression, FINOLA is not the optimal choice for image reconstruction. The strength of the local constraint poses limitations. In the next section, we explore methods to relax this constraint and enhance image reconstruction quality.

## 2.3 RELAXATION OF THE LOCAL CONSTRAINT

In this section, we introduce our second extension of FINOLA, which focuses on relaxing the local constraint to achieve an enhanced solution of the same wave equations for image reconstruction.

**Relaxation through decomposition:** We relax the local constraint by decomposing the solution of one-way wave equations, denoted as $\boldsymbol{\zeta}$, into a collection of special solutions $\boldsymbol{\psi}_i$ that still adhere to the local constraint. This decomposition is expressed as follows:

$$\boldsymbol{\zeta} = \sum_{i=1}^{M}\boldsymbol{\psi}_i, \quad \frac{\partial\boldsymbol{\psi}_i}{\partial x} = \boldsymbol{H}_x\hat{\boldsymbol{\psi}}_i, \quad \frac{\partial\boldsymbol{\psi}_i}{\partial y} = \boldsymbol{H}_y\hat{\boldsymbol{\psi}}_i, \tag{8}$$

where each $\boldsymbol{\psi}_i$ represents a special solution of wave equations (as $\frac{\partial\boldsymbol{\psi}_i}{\partial x} = \boldsymbol{H}_x\boldsymbol{H}_y^{-1}\frac{\partial\boldsymbol{\psi}_i}{\partial y} = \boldsymbol{\Lambda}\frac{\partial\boldsymbol{\psi}_i}{\partial y}$) with local constraint. Due to the linearity of wave equations, $\boldsymbol{\zeta}$ is also a solution of wave equations. However it does not adhere to the local constraint as $\hat{\boldsymbol{\psi}}_i$ involves normalization that is nonlinear. Therefore, after relaxation, the PDEs in FINOLA (Eq. 3) no longer hold, and the one-way wave equations become the new governing equations. It's important to note that the coefficient matrices $\boldsymbol{H}_x$ and $\boldsymbol{H}_y$ are shared by all $\boldsymbol{\psi}_i$.

It's worth noting that this relaxation not only eases the local constraint but also maintains the advantage of computing the entire solution $\boldsymbol{\zeta}(x, y)$ from the initial condition $\boldsymbol{\psi}_i(0, 0)$. Moreover, it demonstrates a significant improvement in reconstruction quality, as confirmed in our experiments. For instance, when employing $C = 2048$ wave equations and decomposing them into 8 special solutions, we observed a significant enhancement in the reconstruction PSNR from 24.8 to 28.0.

**Implementation as multi-path FINOLA:** In practical terms, achieving relaxation can be easily realized by expanding FINOLA from a single path to multiple paths. As depicted in Figure 2, an image undergoes encoding into $M$ vectors, with each vector subjected to the FINOLA process. Each path corresponds to a special solution $\boldsymbol{\psi}_i$ in Eq. 8. Subsequently, the resulting feature maps are aggregated to reconstruct the original image. It's important to note that all these paths share the same set of parameters. Our experiments have validated the effectiveness of this approach in generating feature maps at multiple resolutions, ranging from $16\times16$ to $128\times128$ for image with size $256\times256$.

**Two controlling parameters:** Our method relies on two key controlling parameters: (a) the number of wave equations, denoted as $C$ (or the number of channels), and (b) the number of special solutions,

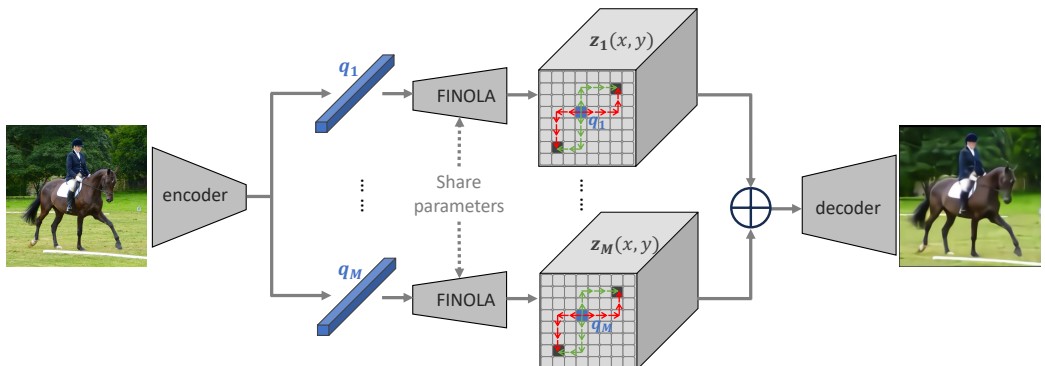

Figure 2: **Multi-path FINOLA:** The input image is encoded into $M$ vectors $\boldsymbol{q}_1, \ldots, \boldsymbol{q}_M$. Then the shared FINOLA is applied on each $\boldsymbol{q}_i$ to generate feature map $\boldsymbol{z}_i(x, y)$ by first-order norm+linear autoregression (see Chen et al. (2023) for details). The aggregated feature map is decoded by applying upsampling and convolution to reconstruct the image. Best viewed in color.

denoted as $M$ in Eq. 8 (or the number of FINOLA paths). Experimental results demonstrate that increasing the value of either parameter leads to reduced error in the reconstruction process.

## 2.4 Revisiting Diagonalizability of $AB^{-1}$

Here, we revisit the diagonalization of $\boldsymbol{AB}^{-1}$ to provide mathematical clarification.

**Wave speeds are not explicitly trainable:** It is crucial to note that wave speeds, denoted as $\boldsymbol{\Lambda}$, are not explicitly learned during training. Instead, they are computed post-training through the diagonalization of trainable matrices $\boldsymbol{A}$ and $\boldsymbol{B}$ (see Eq. 4). A meticulous examination of the eigenvalues in $\boldsymbol{\Lambda}$ and eigenvectors in $\boldsymbol{V}$ across multiple trained models confirms their complex nature.

**Diagonalizability of $AB^{-1}$ is not guaranteed but empirical:** Matrices $\boldsymbol{A}$ and $\boldsymbol{B}$ are learned from training loss without any imposed constraints. Consequently, the diagonalizability of $\boldsymbol{AB}^{-1}$ is not guaranteed, even over complex values $\mathbb{C}$. However, in practice, our experiments indicate that non-diagonalizable matrices rarely occur. This empirical observation suggests that the set of matrices resistant to diagonalization is sufficiently small to escape detection through the learning process.

**Diagonalizability of $AB^{-1}$ indicates separable one-way wave equations:** When $\boldsymbol{AB}^{-1}$ is not diagonalizable, the equation $\frac{\partial \boldsymbol{z}}{\partial x} = \boldsymbol{AB}^{-1} \frac{\partial \boldsymbol{z}}{\partial y}$ in Eq. 4 still holds, resembling a vectorized version of a one-way wave equation with multiple dimensions linearly entangled. In contrast, when $\boldsymbol{AB}^{-1}$ is diagonalizable, the dimensions become disentangled after diagonalization. Consequently, each dimension (after projection by the inverse of the eigenvector matrix) follows a one-dimensional wave equation.

## 2.5 Real-valued Wave Speeds

It is worth noting that the speeds of the wave equations are generally complex numbers $\lambda_k \in \mathbb{C}$, which is also validated in the experiments. This arises because we do not impose constraints on the coefficient matrices ($\boldsymbol{A}$, $\boldsymbol{B}$) in Eq. 4. Consequently, during the diagonalization process, $\boldsymbol{AB}^{-1} = \boldsymbol{V\Lambda V}^{-1}$, it is highly likely that the eigenvalues and eigenvectors will be complex numbers.

Here, we introduce two interesting cases by constraining the speeds of the one-way wave equations as follows: (a) as real numbers $\lambda_k \in \mathbb{R}$, and (b) as all equal to one $\lambda_1 = \cdots = \lambda_C = 1$.

**Real speed $\lambda_k \in \mathbb{R}$:** This is achieved by constraining matrices $\boldsymbol{H}_x$ and $\boldsymbol{H}_y$ in Eq. 6 as real diagonal matrices:

$$\boldsymbol{H}_x = \text{diag}(\alpha_1, \alpha_2, \ldots, \alpha_C), \quad \boldsymbol{H}_y = \text{diag}(\beta_1, \beta_2, \ldots, \beta_C), \quad \boldsymbol{A} = \boldsymbol{PH}_x, \quad \boldsymbol{B} = \boldsymbol{PH}_y. \quad (9)$$

Here, the coefficient matrices $\boldsymbol{A}$ and $\boldsymbol{B}$ in FINOLA are implemented by multiplying a real projection matrix $\boldsymbol{P}$ with diagonal matrices $\boldsymbol{H}_x$ and $\boldsymbol{H}_y$, respectively. Consequently, the speeds of the wave equations are real numbers, denoted as $\lambda_k = \alpha_k / \beta_k$.

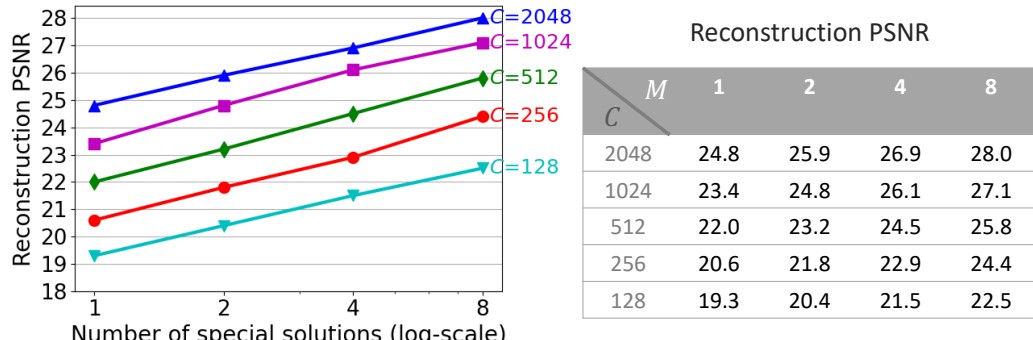

Figure 3: **Reconstruction PSNR vs. number of special solutions** ($M$). Increasing the number of special solutions (as described in Eq. 8) consistently improves reconstruction PSNR across different dimensions ($C = 128$ to $C = 2048$), validating the relaxation of local constraints. The resolution of waves $\zeta$ (or the feature map fed in decoder) is set to $64 \times 64$, and image size is $256 \times 256$. Best viewed in color.

**All-one speed** $\lambda_1 = \cdots = \lambda_C = 1$**:** By further constraining $\boldsymbol{H}_x$ and $\boldsymbol{H}_y$ as identity matrices, all wave equations have identical speed $\lambda_k = 1$.

$$\boldsymbol{H}_x = \boldsymbol{H}_y = \boldsymbol{I}, \quad \boldsymbol{A} = \boldsymbol{B} = \boldsymbol{P}, \quad \lambda_1 = \lambda_2 = \cdots = \lambda_C = 1. \tag{10}$$

Here, the coefficient matrices $\boldsymbol{A}$ and $\boldsymbol{B}$ in FINOLA are also identical and denoted as $\boldsymbol{P}$.

Experimental results (see Figure 7) show that although reconstruction error increases when applying these two constraints, the reconstruction is still reasonably good.

### 2.6 Scattering Initial Conditions Spatially

To enhance reconstruction further, we can mix spatially shifted hidden waves by adjusting spatial positions to place the initial conditions, all without introducing additional parameters or FLOPs. Please refer to Appendix A for the technical details. Implementing this concept is straightforward through multi-path FINOLA (refer to Figure 2), where each path utilizes shared parameters but employs scattered initial positions rather than overlapped at the center. We will show later in experiments that choosing proper initial position results in performance boost.

## 3 Experiments

We assess the performance of hidden waves for image reconstruction on the ImageNet-1K dataset Deng et al. (2009). Our models are trained on the training set and subsequently evaluated on the validation set. For detailed model and training information, please refer to Appendix B. The default wave resolution $\zeta$ (or the size of feature map) is set at $64 \times 64$, and the default image size is $256 \times 256$.

### 3.1 Comparisons with FINOLA

**Hidden waves enhances mathematical description of images by generalizing FINOLA:** Figure 3 illustrates the PSNR values for image reconstruction on the ImageNet-1K validation set. In comparison to vanilla FINOLA (first column, $M = 1$), *Hidden Waves enhances the reconstruction PSNR for the same size of feature map* by aggregating multiple special FINOLA solutions to relax local constraints. For instance, with $C = 2048$ equations, the summation of $M = 8$ special solutions achieves a PSNR of 28.0, significantly surpassing the use of a single $M = 1$ solution (24.8 PSNR). This trend, where larger values of $M$ result in higher PSNR, holds true across various numbers of equations, ranging from $C = 128$ to 2048. Visual comparisons in Figure 5 in Appendix C further highlight that *Hidden Waves* with $M = 8$ special solutions exhibit markedly superior image quality compared to the vanilla FINOLA approach ($M = 1$).

These results demonstrate that *Hidden Waves* enhances the mathematical description of images, i.e., the feature map governed by one-way wave equations is more accurate than the one governed by

Table 1: **Comparing *hidden waves* with FI-NOLA on parameter efficiency:** *hidden waves* achieves comparable reconstruction PSNR by using significant fewer parameters in matrcies $A$ and $B$.

| METHOD | LATENT | DIM of $A/B$ ↓ | PSNR ↑ |
|---|---|---|---|
| FINOLA | 512×1 | 512×512 | **22.0** |
| **Hidden Waves** | 256×2 | **256×256** ($\times\frac{1}{4}$) | 21.8 |
| FINOLA | 1024×1 | 1024×1024 | **23.4** |
| **Hidden Waves** | 256×4 | **256×256** ($\times\frac{1}{16}$) | 22.9 |
| FINOLA | 2048×1 | 2048×2048 | **24.8** |
| **Hidden Waves** | 256×8 | **256×256** ($\times\frac{1}{64}$) | 24.4 |
| FINOLA | 4096×1 | 4096×4096 | **25.9** |
| **Hidden Waves** | 256×16 | **256×256** ($\times\frac{1}{256}$) | 25.4 |

Table 2: **Comparison with convolutional auto-encoder (Conv-AE).** PSNR values for image reconstruction on the ImageNet-1K validation set are reported. *Hidden wave* achieves a higher reconstruction PSNR compared to Conv-AE with the same latent size, while utilizing significantly fewer parameters in the decoder (see the third column). Both methods employ the same Mobile-Former encoder.

| METHOD | LATENT | #PARAM ↓ | PSNR ↑ |
|---|---|---|---|
| Conv-AE | 2048 | 35.9M | 24.6 |
| **Hidden Waves** | 2048$_{(1024\times2)}$ | **16.6M** | **24.8** |
| Conv-AE | 8192 | 35.9M | 26.0 |
| **Hidden Waves** | 8192$_{(1024\times8)}$ | **16.6M** | **27.1** |

the two PDEs in FINOLA. Notably, *Hidden Waves* improves FINOLA elegantly by summing over multiple FINOLA without introducing additional parameters. The feature map, after summation, remains within the solution space of wave equations through linearity but departs from the solution space of FINOLA. We acknowledge that *Hidden Wave* has a larger latent size than FINOLA, explaining why multiple FINOLA paths effortlessly navigate the solution space of wave equations for a more optimal solution. Encoding more initial conditions simplifies the approach toward the optimal solution within the wave equation space.

**Hidden waves is more parameter-efficient than FINOLA:** Furthermore, leveraging multiple paths in FINOLA offers a parameter-efficient approach to enhance reconstruction quality with a larger latent size. In contrast to vanilla FINOLA, which improves reconstruction quality by increasing the number of channels and the sizes of matrices $A$ and $B$, *Hidden Waves* achieves the same by adding paths (or initial conditions) for the same number of wave equations and parameters in matrices $A$ and $B$. Table 1 compares *Hidden Waves* with FINOLA across various latent sizes, ranging from 512 to 4096. Both methods achieve higher PSNR through different strategies for increasing the latent size (adding paths in *Hidden Waves* vs. increasing channels in FINOLA). While *Hidden Waves* trails slightly behind FINOLA in terms of PSNR, it maintains a constant size for matrices $A$ and $B$, which is 256 times smaller than FINOLA at a latent size of 4096. This becomes crucial as the latent size increases. For instance, at a latent size of 16,384, FINOLA requires 268 million parameters in matrices $A$ and $B$, whereas *Hidden Waves* incurs only 4 million parameters (2048x2048) by aggregating 8 FINOLA paths.

## 3.2 COMPARISON WITH PREVIOUS ENCODING-DECODING TECHNIQUES

**Comparisons with convolutional auto-encoder Masci et al. (2011); Ronneberger et al. (2015); Rombach et al. (2021):** Table 2 presents a comparison between our method and convolutional autoencoder (Conv-AE) concerning image reconstruction, measured by PSNR. Both approaches share the same Mobile-Former Chen et al. (2022) encoder and have identical latent sizes (2048 or 8192). In our method, multi-path FINOLA is initially employed to generate a 64×64 feature map, followed by a convolutional decoder to reconstruct an image with a size of 256×256. On the other hand, Conv-AE employs a deeper decoder that utilizes convolution and upsampling from the latent vector to reconstruct an image. *Hidden wave* has significantly fewer parameters in the decoder. The results highlight the superior performance of our method over Conv-AE, indicating that a single-layer *hidden waves* is more effective than a multi-layer convolution and upsampling approach.

**Comparisons with discrete cosine transform (DCT) Ahmed et al. (1974):** Table 3-(a) compares Hidden Wave with DCT. DCT is conducted per 8×8 image block, and the top-left $K$ coefficients (in zig-zag manner) are kept, while the rest are set to zero. We choose four $K$ values (1, 3, 6, 10) for comparison. Clearly, *hidden waves* achieves a higher PSNR with a smaller latent size.

**Comparisons with discrete wavelet transform (DWT/DTCWT) Strang (1989); Daubechies (1992); Vetterli & Kovacevic (2013):** We compare *hidden waves* with DWT and DTCWT in Table 3-(a). Three scales are chosen for wavelet decomposition. The comparisons are organized into three

Table 3: **Comparison with discrete cosine transform (DCT) and discrete wavelet transform (DWT).** PSNR values of image reconstruction are reported on the ImageNet-1K validation set for different latent size. $^\dagger$ refers to placing 8 initial conditions at different positions rather than overlapping at the center.

| METHOD | LATENT ↓ | PSNR ↑ |
|---|---|---|
| DCT (top-left 1) | 3072 | 20.6 |
| **Hidden Waves** | **2048** $_{(1024 \times 2)}$ | **24.8** |
| DCT (top-left 3) | 9216 | 23.5 |
| **Hidden Waves** | **8192** $_{(1024 \times 8)}$ | **27.1** |
| DCT (top-left 6) | 18432 | 25.6 |
| **Hidden Waves** | **16384** $_{(2048 \times 8)}$ | **28.0** |
| DCT (top-left 10) | 30720 | 27.5 |
| **Hidden Waves** | **16384** $^\dagger_{(2048 \times 8)}$ | **28.9** |

(a) **Comparison with DCT.**

| METHOD | LATENT ↓ | PSNR ↑ |
|---|---|---|
| DWT (scale-3 LL subband) | 3888 | 21.5 |
| DTCWT (scale-3 LL subband) | 12288 | 22.3 |
| **Hidden Waves** | **2048** $_{(1024 \times 2)}$ | **24.8** |
| DWT (scale-3 all subbands) | 15552 | 24.3 |
| DTCWT (scale-3 all subbands) | 49152 | 25.6 |
| **Hidden Waves** | **8192** $_{(1024 \times 8)}$ | **27.1** |
| DWT (scale-2 all subbands) | 55953 | 28.7 |
| DTCWT (scale-2 all subbands) | 196608 | **30.8** |
| **Hidden Waves** | **16384** $^\dagger_{(2048 \times 8)}$ | 28.9 |

(b) **Comparison with DWT/DTCWT.**

Table 4: **Validation across multiple resolutions and image sizes:** PSNR values of image reconstruction are reported for multiple resolutions and image sizes on the ImageNet-1K validation set. $C = 1024$ for the number of wave equations (or feature map dimension), and $M = 4$ for the number of special solutions. Default resolution and image size are indicated with $^\dagger$.

| RESOLUTION | 16×16 | 32×32 | 64×64$^\dagger$ | 128×128 | IMAGE SIZE | 256×256$^\dagger$ | 512×512 |
|---|---|---|---|---|---|---|---|
| PSNR | 26.2 | 26.2 | 26.1 | 25.4 | PSNR | 26.1 | 25.1 |

(a) **Resolution of feature map (or solution of wave equations).**  (b) **Image size.**

groups: (a) using only the LL subband at the coarsest scale (scale 3), (b) using all subbands (LL, LH, HL, HH) at the coarsest level, and (c) using all subbands at the finer scale (scale 2). *Hidden waves* outperforms DWT and DTCWT in terms of PSNR for the first two groups, achieving at a smaller latent size. In the last group, while *hidden-wave*'s PSNR is lower than DTCWT, its latent size is significantly smaller (more than 10 times smaller).

## 3.3 MAIN PROPERTIES

We ablate our *hidden waves* using the default setting as follows: utilizing $C = 1024$ one-way equations (equivalent to 1024 channels), each represented by aggregating $M = 4$ special solutions (refer to Eq. 8). The default wave resolution $\zeta$ (or the size of feature map) is set at 64×64, and the default image size is 256×256.

**Hidden Waves perform consistently well across multiple resolutions:** Table 4-(a) displays reconstruction PSNR scores across different wave resolutions (equivalent to feature map resolutions). The reconstruction accuracy remains consistent across various resolutions, with slightly reduced performance at 128×128. This decrease is primarily attributed to the lighter decoder (see Table 8 in Appendix B). Specifically, the decoder at 128×128 resolution (1.9M parameters) is only 40% the size of the 64×64 resolution decoder (4.7M parameters).

Table 4-(b) presents results for image sizes of 256×256 and 512×512. Hidden waves also perform well on larger images, albeit with slightly lower PSNR scores compared to smaller images. This reduction in PSNR is attributed to the higher compression rate of the encoder. In both cases, the encoder outputs maintain a consistent size of 1024×4. We intentionally maintain this dimension to assess the model's performance in handling larger images with more visual details to encode.

These results underscore the capability of our hidden waves method to reconstruct images at various resolutions and image sizes, affirming the validity of this novel mathematical approach.

**Performance comparison of complex and real-valued speeds:** Table 5 illustrates the effects of employing various number types for wave speeds. Corresponding visual examples of reconstructions are provided in Figure 7 (Appendix C). Compared to the default scenario with complex-valued wave speeds, two modifications, i.e. enforcing wave speeds as real numbers or uniformly setting them to one (see Section 2.5), result in a slight decline in performance. Nevertheless, both real-valued cases still exhibit reasonably good reconstruction quality.

Table 5: **Wave speed types and reconstruction accuracy.** PSNR values for image reconstruction on the ImageNet-1K validation set are reported. Complex-valued wave speeds yield more accurate reconstruction compared to real-valued speeds.

| WAVE SPEED | $M$ | $C$ | PSNR $\uparrow$ |
|---|---|---|---|
| Complex $\lambda_k \in \mathbb{C}$ | 4 | 1024 | **26.1** |
| Real $\lambda_k \in \mathbb{R}$ | 4 | 1024 | 25.1 |
| All-one $\lambda_k = 1$ | 4 | 1024 | 23.9 |

Table 6: **Position of initial conditions.** PSNR values for image reconstruction on the ImageNet-1K validation set is reported. Scattering of initial positions spatially boosts performance.

| POSITION | $M$ | $C$ | PSNR $\uparrow$ |
|---|---|---|---|
| Overlapping at Center | 4 | 1024 | 26.1 |
| Scattering Uniformly | 4 | 1024 | **26.7** |
| Overlapping at Center | 8 | 1024 | 27.1 |
| Scattering Uniformly | 8 | 1024 | **28.0** |
| Overlapping at Center | 8 | 2048 | 28.0 |
| Scattering Uniformly | 8 | 2048 | **28.9** |

**Position of initial conditions:** Table 6 demonstrates that further improvements in reconstruction can be achieved by selecting appropriate positions for the initial conditions. Scattering the initial conditions uniformly yields superior results compared to the central overlap, regardless of whether we use 4 or 8 special FINOLA solutions.

## 4  RELATED WORKS

**Image autoregression:** Autoregression has played a pivotal role in generating high-quality images van den Oord et al. (2016b;a); Salimans et al. (2017); Chen et al. (2018b). These methods model conditional probability distributions of current pixels based on previously generated ones, evolving from pixel-level focus to latent space modeling using vector quantization van den Oord et al. (2017); Razavi et al. (2019); Esser et al. (2021); Yu et al. (2022). Notably, FINOLA Chen et al. (2023) has recently introduced a concise mathematical format, demonstrating that images follow a first-order, norm+linear autoregressive pattern in a deterministic manner. In our approach, we reveals new insights by generalizing FINOLA as a set of one-way wave equations and enhancing image reconstruction through the relaxation of local constraints.

**Image transforms:** The Discrete Cosine Transform (DCT) Ahmed et al. (1974) and Wavelet Transform Strang (1989); Daubechies (1992); Vetterli & Kovacevic (2013) are widely recognized signal processing techniques for image compression. The DCT transforms images into a frequency domain and allows for efficient quantization and encoding of image coefficients. On the other hand, the Wavelet Transform is a versatile technique that decomposes signals into different scales and frequencies, enabling both spatial and frequency domain analysis. Both DCT and wavelet transforms project images into a *complete* space consisting of *known* wave functions, in which each image has *compact* coefficients, i.e., most coefficients are close to zero. In contrast, our method offers a distinct mathematical perspective for representing images. It encodes images into a *compact* space represented by a set of *one-wave equations* with *hidden* speeds. Each image corresponds to a unique set of initial conditions. These differences are summarized in Table 13 at Appendix E.

## 5  CONCLUSION

In this paper, we've unveiled the intriguing *hidden-waves* phenomenon. It enables the successful image reconstruction using one-way wave equations with hidden and learnable speeds. Each image corresponds to a solution with a unique initial condition, computable via a visual encoder. Furthermore, our exploration has revealed two critical mathematical properties. Firstly, the solution for each image can be elegantly decomposed into a collection of special solutions, all governed by the same one-way wave equations. These special solutions possess the coveted attribute of being first-order autoregressive, and they share coefficient matrices for autoregression. Secondly, we have identified that the product of the coefficient matrices forms a diagonal matrix, with wave speeds as diagonal elements. This discovery demonstrates that the speeds of waves and autoregression coefficients remain consistent and invariant across diverse images, transcending their individual content. The *hidden-waves* phenomenon provides a unique mathematical insight that extends beyond the realm of individual images.

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

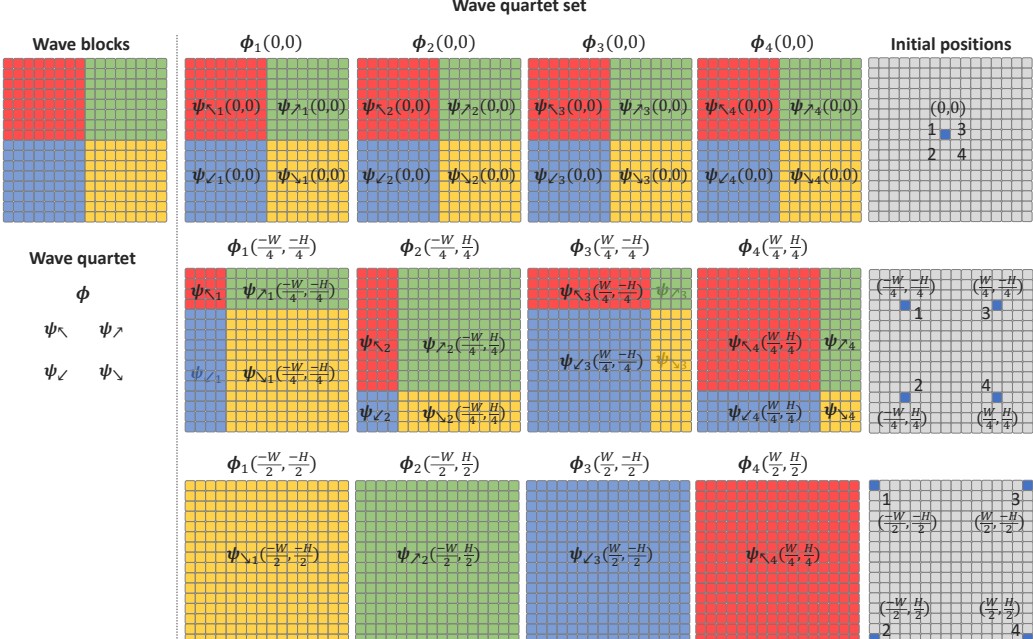

Figure 4: **Wave quartet with different initial positions.** (Left) Illustration of wave blocks and their corresponding wave quartet. (Right) Three examples of wave quartet sets with different initial condition positions. The first row shows all four wave quartets sharing the initial position at the center $(0, 0)$. The last row depicts the initial positions at the four corners, resulting in only one wave block being available. The second row represents an intermediate scenario where each wave quartet corresponds to four wave blocks with varying sizes. Best viewed in color.

## A  MIXTURE OF SPATIALLY SHIFTED HIDDEN WAVES

To enhance reconstruction further, we can mix spatially shifted hidden waves by adjusting the initial condition positions, all without introducing additional parameters or FLOPs. Below, we delve into the technical details.

**Wave block:** When implementing a special solution $\psi_i$ using FINOLA, it inherently involves four sets of wave equations instead of one. This arises from FINOLA's use of additional matrices $\boldsymbol{A}_-$ and $\boldsymbol{B}_-$ to handle autoregression in left and upward directions. This configuration results in four sets of one-way wave equations, one for each quadrant block when the initial condition is at the center of the feature map. The first row of Figure 4 illustrate the four quadant blocks by differnt colors. We refer to this as the *wave block*. The wave speeds for each quadrant can be computed in a manner similar to Eq. 4 and 5. For instance, the wave speeds of the top-left quadrant correspond to the eigenvalues of $\boldsymbol{A}_- \boldsymbol{B}_-^{-1}$.

**Wave quartet:** We group the solutions of the four wave quadrant blocks (each representing a set of one-way wave equations) into a single set denoted as $\boldsymbol{\phi} = \{\psi_\nwarrow, \psi_\nearrow, \psi_\swarrow, \psi_\searrow\}$, where the arrow indicates the direction of autoregression for the corresponding quadrant. For example, $\psi_\nwarrow$ represents a special solution for the top-left quadrant. We refer to this set as the *wave quartet*.

**Position of initial condition:** The wave quartet shares a common initial condition, which is typically located at the center of the feature map and denoted as $(0, 0)$. When this initial condition is moved to different positions, it leads to variations in the sizes of the four wave blocks (as illustrated in the second row of Figure 4). We incorporate the position of the initial condition into the wave quartet representation as $\boldsymbol{\phi}(u, v) = \psi_\nwarrow(u, v), \psi_\nearrow(u, v), \psi_\swarrow(u, v), \psi_\searrow(u, v)$, where $(u, v)$ indicates the position of the initial condition.

**Mixture of shifted hidden waves:** Similar to Eq. 8 that aggregates a set of special solutions, we add multiple wave quartets that are initialized at different positions. The wave quartets with distinct initial positions results in different layouts of wave blocks. Figure 4 illustrates three cases: the first

Table 7: **Specification of Mobile-Former encoder**. "bneck-lite" denotes the lite bottleneck block Li et al. (2021). "M-F" denotes the Mobile-Former block and "M-F$^{\downarrow}$" denotes the Mobile-Former block for downsampling.

| STAGE | RESOLUTION | BLOCK | #EXP | #OUT |
|:---:|:---:|:---:|:---:|:---:|
| stem | $256^2$ | conv 3×3 | – | 64 |
| 1 | $128^2$ | bneck-lite | 128 | 64 |
| 2 | $64^2$ | M-F$^{\downarrow}$ | 384 | 112 |
| | | M-F | 336 | 112 |
| 3 | $32^2$ | M-F$^{\downarrow}$ | 672 | 192 |
| | | M-F | 576 | 192 |
| | | M-F | 576 | 192 |
| 4 | $16^2$ | M-F$^{\downarrow}$ | 1152 | 352 |
| | | M-F | 1408 | 352 |
| | | M-F | 1408 | 352 |
| | | M-F | 2112 | 480 |
| | | M-F | 2880 | 480 |
| | | M-F | 2880 | 480 |
| | | conv 1×1 | – | 2880 |

row has all four wave quartets sharing the initial position at the center $(0,0)$, while the last row has the initial positions at the four corners. When the initial position is located at a corner, only one wave block is available. The second row represents an intermediate scenario where each wave quartet corresponds to four wave blocks with varying sizes.

Implementing this concept is straightforward through multi-path FINOLA (refer to Figure 2), where each path utilizes shared parameters but employs different initial positions. As different paths share parameters, multiple wave quartets share the four sets of one-way wave equations. The addition of wave quartets effectively partitions the entire feature map into regions, each combining multiple wave blocks differently. We will show later in experiments that choosing proper initial position results in performance boost.

## B    Network Architectures and Training Setup

In this section, detailed information on the network architecture used in our study is provided. Specifically, we describe (a) the Mobile-Former encoders and (b) the covolutional decoders.

**Mobile-Former encoder:** Mobile-Former Chen et al. (2022) is used as the encoder in our study, which is detailed in Table 7. It is a CNN-based network that extends MobileNet Sandler et al. (2018) by adding 6 global tokens in parallel. To preserve spatial details, we increase the resolution of the last stage from $\frac{1}{32}$ to $\frac{1}{16}$.

**Decoders:** The architecture details of the decoders are presented in Table 8. As the hidden wave spatial resolution increases from 16×16 to 128×128), the decoder's complexity decreases with fewer upsampling and convolution blocks.

**Training setup:**

The training settings for using hidden waves to reconstruct images are provided in Table 9. The learning rate is scaled as $lr = base\_lr \times$batchsize / 256.

## C    Visualization

**Comparison between Hidden-Wave and FINOLA:** Figure 5 visually demonstrates that hidden waves with $M = 8$ special solutions exhibit markedly superior image quality compared to the vanilla FINOLA approach ($M = 1$).

**Reconstruction examples for varying number of wave equations and special solutions:** Figure 6 illustrates the reconstruction examples obtained for different combinations of one-way equations

Table 8: **Decoder specifications**. The decoder's complexity decreases as the hidden wave spatial resolution increases from 16×16 to 128×128). "res-conv" represents a residual block He et al. (2016) consisting of two 3x3 convolutional layers, while "up-conv" performs upsampling followed by a 3x3 convolutional layer.

| resolution | 16×16 | | | 32×32 | | | 64×64 | | | 128×128 | | |
|---|---|---|---|---|---|---|---|---|---|---|---|---|
| | block | #block | #out | block | #block | #out | block | #block | #out | block | #block | #out |
| $16^2$ | res-conv | 1 | 512 | | | | | | | | | |
| $32^2$ | up-conv | 1 | 512 | | | | | | | | | |
| | res-conv | 1 | 256 | res-conv | 1 | 256 | | | | | | |
| $64^2$ | up-conv | 1 | 256 | up-conv | 1 | 256 | | | | | | |
| | res-conv | 1 | 256 | res-conv | 1 | 256 | res-conv | 1 | 256 | | | |
| $128^2$ | up-conv | 1 | 256 | up-conv | 1 | 256 | up-conv | 1 | 256 | | | |
| | res-conv | 1 | 128 | res-conv | 1 | 128 | res-conv | 1 | 128 | res-conv | 1 | 128 |
| $256^2$ | up-conv | 1 | 128 | up-conv | 1 | 128 | up-conv | 1 | 128 | up-conv | 1 | 128 |
| | res-conv | 1 | 128 | res-conv | 1 | 128 | res-conv | 1 | 128 | res-conv | 1 | 128 |
| | conv3×3 | 1 | 3 | conv3×3 | 1 | 3 | conv3×3 | 1 | 3 | conv3×3 | 1 | 3 |

Table 9: **Training setting for hidden waves.**

| CONFIG | VALUE |
|---|---|
| optimizer | AdamW |
| base learning rate | 1.5e-4 |
| weight decay | 0.1 |
| batch size | 128 |
| learning rate schedule | cosine decay |
| warmup epochs | 10 |
| training epochs | 100 |
| image size | $256^2$ |
| augmentation | RandomResizeCrop |

or channel counts ($C = 128, 256, 512, 1024, 2048$) and the number of special solutions ($M = 1, 2, 4, 8$). These results correspond to the experiments in Figure 3, as discussed in Section 3.1.

Notably, a consistent trend emerges where increasing the value of $M$ consistently enhances image quality. This trend remains consistent across various equation counts, ranging from $C = 128$ to $2048$. This observation underscores the efficacy of relaxing the local constraint, as detailed in Section 2.3, to address the one-way wave equations.

**Reconstruction examples for complex-valued and real-valued wave speeds:** Table 10-(a) provides the results for the two special cases detailed in Section 2.5, while Figure 7 displays corresponding reconstruction examples. In comparison to the default scenario using complex-valued wave speeds, two modifications - enforcing wave speeds as real numbers or setting them uniformly to one - show a slight decline in performance. Nonetheless, both special cases still deliver reasonably good PSNR scores. Notably, the all-one wave speed configuration achieves a PSNR of 23.9. This specific configuration shares the coefficient matrix for autoregression across all four directions, creating symmetry in the feature map. To account for this symmetry, we introduced position embedding before entering the decoder.

In an effort to determine whether position embedding is the dominant factor behind this improvement, we conducted experiments by generating feature maps using both repetition and position embedding, even with three times more channels (3072). However, this approach still falls short of the all-one wave speed configuration by 2.7 PSNR (as detailed in Table 10-(b)). Its reconstruction quality significantly lags behind that of all-one waves, as depicted in the last two columns of Figure 7.

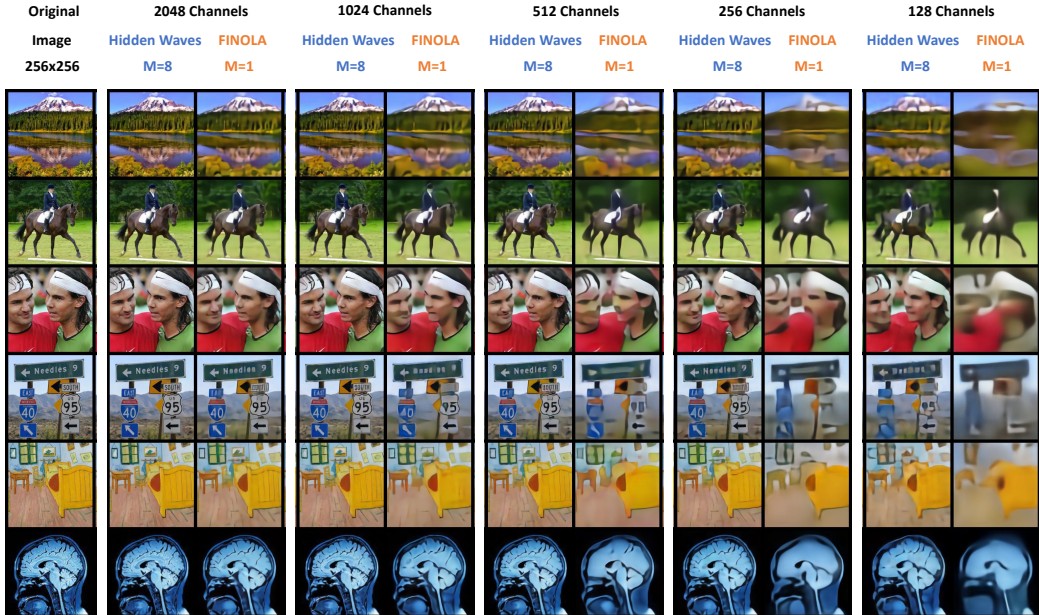

Figure 5: **Hidden waves vs. FINOLA:** Summing $M = 8$ special solutions $\psi_i$ (as in Eq. 8) in hidden waves yields superior image reconstruction quality compared to FINOLA, which employs a single ($M = 1$) special solution. This trend holds across various dimensions (from $C = 128$ to $C = 2048$). Resolution of waves $\zeta$ is set to $64\times64$, with an image size of $256\times256$. Best viewed in color.

## D  MORE EXPERIMENTAL RESULTS

**Plateau onset in PSNR with increasing $M$:** Table 11 extends the number of special FINOLA solutions to $M = 16$, investigate potential improvements. In comparison to the observed rate of change for smaller $M$ values ($1\rightarrow2$, $2\rightarrow4$, $4\rightarrow8$), the PSNR increase from $M = 8$ to $M = 16$ shows a slowdown, indicating the onset of a plateau.

**Position of initial conditions:** Table 12 delves into a more detailed examination of the initial condition positions. When employing four wave quartets, positioning the initial conditions at coordinates such as $(-\frac{W}{4}, -\frac{H}{4})$, $(-\frac{W}{4}, \frac{H}{4})$, $(\frac{W}{4}, -\frac{H}{4})$, and $(\frac{W}{4}, \frac{H}{4})$ results in a remarkable 0.6 PSNR improvement over placing them at the center, denoted as $(0, 0)$.

## E  COMPARISON WITH DCT/WAVELET TRANSFORM

Both DCT and wavelet transforms project images into a *complete* space consisting of *known* wave functions, in which each image has *compact* coefficients, i.e., most coefficients are close to zero. In contrast, our method offers a distinct mathematical perspective for representing images. It encodes images into a *compact* space represented by a set of *one-wave equations* with *hidden* speeds. Each image corresponds to a unique set of initial conditions. These differences are summarized in Table 13.

## F  FUTURE WORK

In future investigations within the context of wave equations, exploring a non-FINOLA-based solution to the wave equation stands out as a key direction, offering potential insights into the dynamics of these equations. Additionally, understanding the relationships between multiple initial conditions in Hidden Waves holds promise, shedding light on their interactions and influence on the model's behavior. Further, exploring the distribution of real and noise images within Hidden Waves' la-

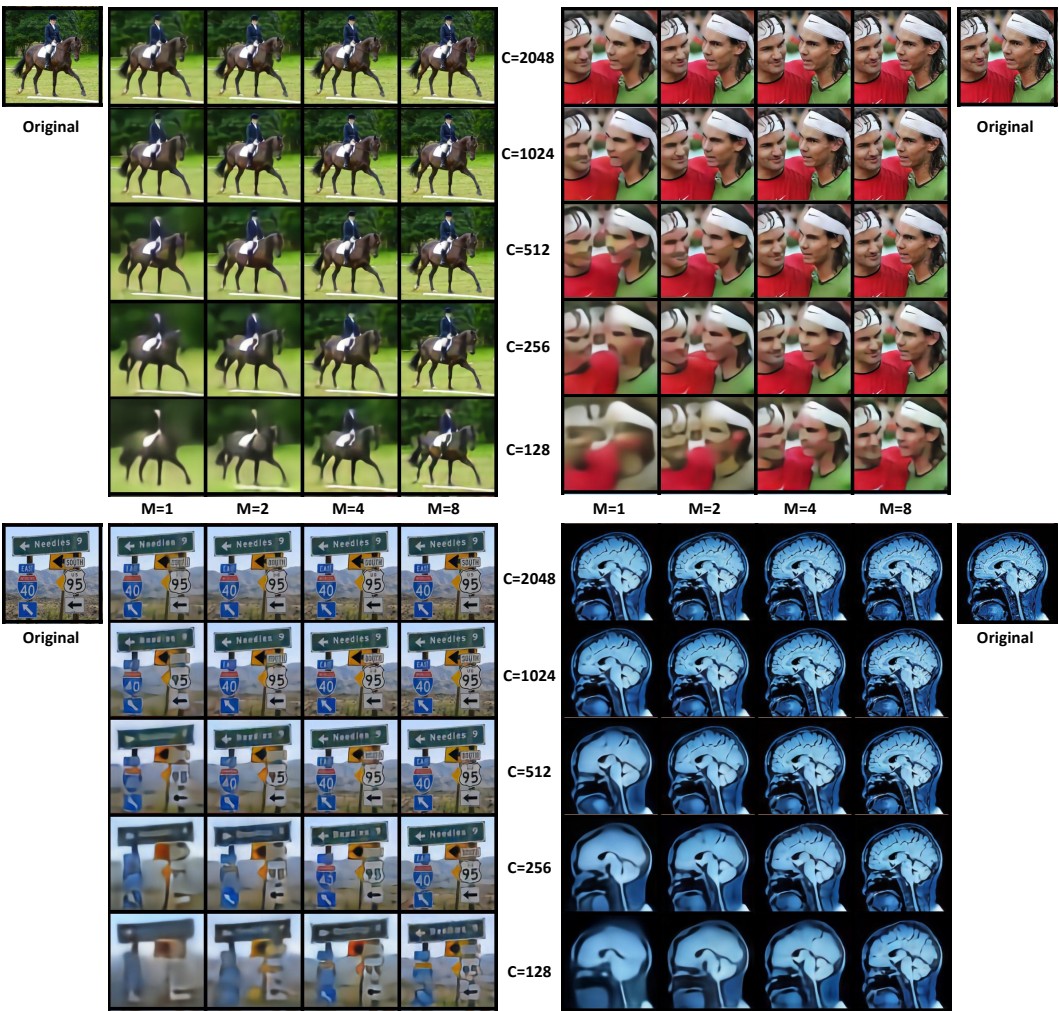

Figure 6: **Reconstruction examples for varying numbers of wave equations ($C$) and special solutions ($M$):** Increasing the number of special solutions, as per Eq. 8, consistently enhances image quality across different dimensions ($C = 128$ to $C = 2048$), affirming the relaxation of local constraints. Wave resolution ($\zeta$) is 64×64, and image size is 256×256. Best viewed in color.

tent space can enhance our understanding of its representation capabilities. Our focus will extend to investigating Hidden Waves' applicability in self-supervised learning and exploring its potential utility in image compression, aiming to efficiently represent and reconstruct images while minimizing storage requirements. These perspectives collectively contribute to advancing Hidden Waves and broadening its applications in deep learning.

Table 10: **Inspection of special cases:** (a) PSNR values for image reconstruction with varying wave speeds (complex, real, all-one) on the ImageNet-1K validation set, with the symbol ‡ denoting the use of position embedding. The number of wave equations (or feature map dimension) is set $C = 1024$, and the number of special solutions is set $M = 4$. (b) A comparison between all-one waves and feature map generation through repetition with position embedding to ensure position embedding isn't the sole dominant factor.

| WAVE SPEED | DIM | PSNR |
|---|---|---|
| Complex $\lambda_k \in \mathbb{C}$ | 1024 | 26.1 |
| Real $\lambda_k \in \mathbb{R}$ | 1024 | 25.1 |
| All-one $\lambda_k = 1^{‡}$ | 1024 | 23.9 |

(a) **Special cases: real and all-one speeds.**

| FEATURE MAP GEN | DIM | PSNR |
|---|---|---|
| Repetition | 3072 | 21.2 |
| All-one waves | **1024** | **23.9** |

(b) **Using position embedding.**

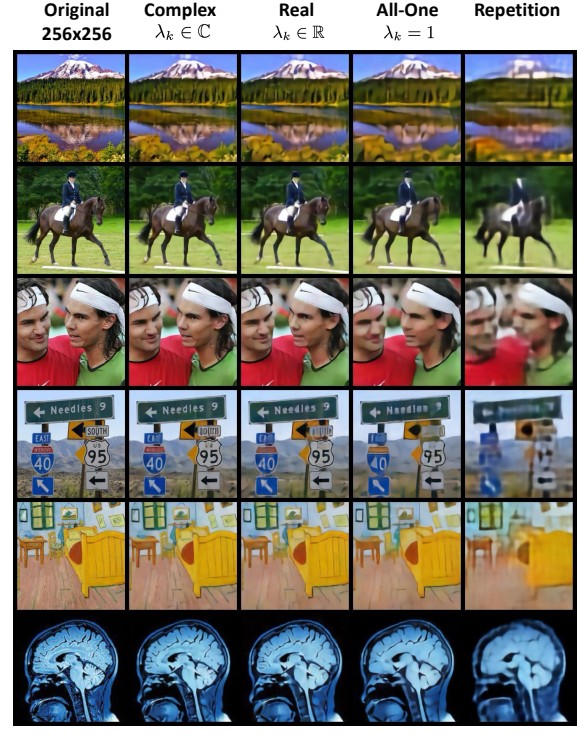

Figure 7: Reconstructed examples for varying wave speeds (complex, real, all-one).

Table 11: **Onset of a plateau at $M = 16$.** In comparison to the rate of change observed over smaller $M$ values ($1\rightarrow2$, $2\rightarrow4$, $4\rightarrow8$), the PSNR increase from $M = 8$ to $M = 16$ slows down, indicating the onset of a plateau.

| CHANNELS | $M = 1$ | $M = 2$ | $M = 4$ | $M = 8$ | $M = 16$ |
|---|---|---|---|---|---|
| $C = 512$ | 22.0 | 23.2 | 24.5 | 25.8 | **26.3** |
| $C = 256$ | 20.6 | 21.8 | 22.9 | 24.4 | **25.5** |
| $C = 128$ | 19.3 | 20.4 | 21.5 | 22.5 | **23.1** |

Table 12: **Position of initial conditions.** PSNR values for image reconstruction on the ImageNet-1K validation set is reported. $C = 1024$ for the number of wave equations (or feature map dimension), and $M = 4$ for the number of wave quartets. $(0, 0)$ denotes all four wave quartets sharing the initial position at the center. $(\pm\frac{W}{2}, \pm\frac{H}{2})$ has the initial positions at the four corners.

| POSITIONS | $(0,0)$ | $(\pm\frac{W}{8}, \pm\frac{H}{8})$ | $(\pm\frac{W}{4}, \pm\frac{H}{4})$ | $(\pm\frac{3W}{8}, \pm\frac{3H}{8})$ | $(\pm\frac{W}{2}, \pm\frac{H}{2})$ |
|---|---|---|---|---|---|
| **PSNR** | 26.1 | 26.2 | **26.7** | 25.5 | 25.2 |

Table 13: Comparison between DCT/Wavelet transform and hidden waves.

| ASPECT | DCT or WAVELET TRANSFORM | HIDDEN WAVES |
|---|---|---|
| Representation of Waves | Cosine/Wavelet functions with *known* parameters | One-way wave equations with *hidden* speeds |
| Individual Image | Unique *coefficients* per image | Unique *initial conditions* per image |
| Compactness | Compact *coefficients* within each image | Compact *space* representation |

