# OpenReview forum: "On the Hidden Waves of Image"
_ICLR.cc/2024/Conference — Submitted to ICLR 2024_

### Official Review · Reviewer_Y4BM · 2023-10-29

**Soundness:** 3 good
**Presentation:** 3 good
**Contribution:** 2 fair
**Rating:** 6
**Confidence:** 3

**Summary:**

This paper introduces the concept of "hidden waves," a phenomenon that enables the successful reconstruction of images through a series of one-way wave equations with hidden and adaptable speeds. To compute the initial conditions for each image, a visual encoder is employed based on the original image. The paper extends the existing framework of FINOLA by making two key enhancements: generalizing it and relaxing the local constraints. These relaxations not only enhance the image reconstruction performance but also lead to the observation of specific phenomena under certain conditions within the representation. They performed rigorous experiments to gain understanding of this perspective.

**Strengths:**

- Through mathematical enhancements and experiments, they develop a deep understanding of FINOLA, which is a general wave representation of an image.
- The section comparing "hidden wave representation" with image autoregression or transformations is well-written and provides clear insights into this encoding.
- They conducted numerous experiments to validate the improvements they implemented.

**Weaknesses:**

- Experimental design

Drawing conclusions from the current results is uncertain due to the limited number of data points and the lack of significant differences. For instance, the interpretation of Table 3 remains unclear. Should we consider choosing initial condition positions different from the center? Overall, experiments were conducted from various perspectives, they do not yield meaningful insights or conclusions.

- The phenomenon is undoubtedly intriguing and interesting, it remains unclear how this method can be applied to future research or practical applications. The paper would be improved by including discussions on potential future perspectives.

**Questions:**

- Were all images resized to the same aspect ratio?  The aspect ratio can impact wave speed, and it may be more natural to preserve the aspect ratio from the original images.
- Conventional image compression methods like DCT transform the image into a linear composition of orthogonal basis. This is simple and provides a straightforward understanding of each coefficient. The paper effectively describes the differences, but what are the main strengths and weaknesses of this method?
- Is the selection of hyperparameters for training this method not very sensitive? How long does the training process take?

Minor comments:
- Use ` for opening quotation mark in latex. Currently ' ... '.
- There are some missing brackets for citations. For example, ... known as FINOLA Chen et al. (2023).

---

> ### Author Response · Authors · 2023-11-20
> **Authors' Rebuttal (part 1)**
>
> Thank you for dedicating your time and effort to provide feedback on our work. Below, we answer the questions that have been raised.
>
> ---
>
> **[Weakness 1] Experimental design: Drawing conclusions from the current results is uncertain due to the limited number of data points and the lack of significant differences. For instance, the interpretation of Table 3 remains unclear. Should we consider choosing initial condition positions different from the center? Overall, experiments were conducted from various perspectives, they do not yield meaningful insights or conclusions.**
>
> Thanks for this constructive suggestion. We will reorganize the experiment section and showcase two key conclusions and three main ablations as follows:
>
> ***Conclusion I: Hidden-Wave Enhances Mathematical Decription.***
> Hidden-Wave demonstrates a superior mathematical description by employing one-way wave equations, resulting in more accurate reconstructions compared to the two PDEs in FINOLA. Figure 4 in the paper supports this conclusion.
>
> ***Conclusion II: Hidden-Wave Outperforms Conventional Methods.***
>
> *Comparison with discrete cosine transform (DCT):*
> The table below showcases the superiority of Hidden-Wave over DCT. DCT operates per 8x8 image block, retaining the top-left $K$ coefficients (in zig-zag manner), with the rest set to zero. Compared to DCT with various $K$ values (1, 3, 6, 10), Hidden-Wave consistently achieves higher PSNR with a smaller latent size.
>
> |Method|Latent Size&darr;|PSNR&uarr;|
> |---|---|---|
> |DCT (top-left 1) | 3072 (32x32x3) | 20.6 |
> | **Hidden Wave (our)** | **2048 (1024x2)** | **24.8** |
> ||||
> |DCT (top-left 3)| 9216 (32x32x9) | 23.5|
> | **Hidden Wave (our)** | **8192 (1024x8)** | **27.1** |
> ||||
> |DCT (top-left 6)| 18432 (32x32x18) | 25.6|
> | **Hidden Wave (our)** | **16384 (2048x8)** | **27.8** |
> ||||
> |DCT (top-left 10)| 30720 (32x32x30) | 27.5|
> | **Hidden Wave (our)** | **16384 (2048x8 mix)** | **28.9** |
>
> Notably, in the last row, the term *"mix"* denotes placing 8 initial conditions at different positions, as opposed to overlapping at the center.
>
> *Comparison with discrete wavelet transform (DWT):*
> We compare Hidden-Wave with DWT in the table below. Three scales are chosen for wavelet decomposition. The comparisons are organized into three groups: (a) using only the LL subband at the coarsest scale (scale 3), (b) using all subbands (LL, LH, HL, HH) at the coarsest level, and (c) using all subbands at the finer scale (scale 2). Hidden-Wave achieves higher PSNR with a smaller latent size.
>
> |Method|Latent Size&darr;|PSNR&uarr;|
> |---|---|---|
> |DWT (scale-3 LL) | 3888 | 21.5 |
> | **Hidden Wave (our)** | **2048 (1024x2)** | **24.8** |
> ||||
> |DWT (scale-3 LL+LH+HL+HH)| 15552 | 24.3|
> | **Hidden Wave (our)** | **8192 (1024x8)** | **27.1** |
> ||||
> |DWT (scale-2 LL+LH+HL+HH)| 55953 | 28.7|
> | **Hidden Wave (our)** | **16384 (2048x8 mix)** | **28.9** |
>
>
> ***Ablation I: Hidden-Wave Consistency across Feature resolutions and Image Sizes.*** Table 1 in the paper supports the claim that Hidden-Wave consistently performs well across multiple feature resolutions and image sizes.
>
> ***Ablation II: Complex-valued Wave Speeds Improve Accuracy.*** Table 2 in the paper indicates that complex-valued wave speeds yield better accuracy compared to real-valued wave speeds.
>
> ***Ablation III: Scattering Initial Conditions Enhance Performance.*** The table below demonstrates that scattering initial condition positions on a circle consistently outperforms overlapping at the center.
>
> |Position|\#Init Conditions ($M$)|\#Channels ($C$)|PSNR&uarr;|
> |---|---|---|---|
> |Overlapping at Center | 4 | 1024 | 26.1 |
> |Scattering on a Circle | 4 | 1024 | **26.7** |
> ||||
> |Overlapping at Center | 8 | 1024 | 27.1 |
> |Scattering on a Circle | 8 | 1024 | **28.0** |
> ||||
> |Overlapping at Center | 8 | 2048 | 27.8 |
> |Scattering on a Circle | 8 | 2048 | **28.9** |
>
> This reorganization highlights the key conclusions and ablations, providing a clearer and more structured presentation of the experimental findings.
>
> ---
>
> **[Weakness 2]: The phenomenon is undoubtedly intriguing and interesting, it remains unclear how this method can be applied to future research or practical applications. The paper would be improved by including discussions on potential future perspectives.**
>
> Excellent point! We will include discussions on potential future perspectives, encompassing (a) understanding the relationship between multiple initial conditions, (b) exploring the distribution of real/noise images in the latent space of Hidden-Wave, (c) investigating the application of Hidden-Wave in self-supervised learning, and (d) exploring the potential use of Hidden-Wave in image compression.

---

> > ### Author Response · Authors · 2023-11-20
> > **Authors' Rebuttal (part 2)**
> >
> > **[Q1]: Were all images resized to the same aspect ratio? The aspect ratio can impact wave speed, and it may be more natural to preserve the aspect ratio from the original images.**
> >
> > During the training, the aspect ratio is randomly set between 0.75 and 1.33. During inference, all images are resized to an aspect ratio of 1:1. We also experimented with maintaining the original aspect ratio during inference, and the difference in PSNR was found to be negligible.
> >
> > When changing the aspect ratio for an input image during inference, although the learned wave speeds are fixed, the contribution of these waves (with different speeds) changes accordingly. This change is reflected in the initial condition. Thus, all waves collaborate around the new initial condition to successfully reconstruct the image with the new aspect ratio.
> >
> > ---
> >
> > **[Q2]: Conventional image compression methods like DCT transform the image into a linear composition of orthogonal basis. This is simple and provides a straightforward understanding of each coefficient. The paper effectively describes the differences, but what are the main strengths and weaknesses of this method?**
> >
> > Great question. Below, we discuss the strengths and weaknesses of Hidden-Wave, compared with conventional image compression methods like DCT transform.
> >
> > ***Strength I: Concise Mathematical Insight.***
> > Hidden-Wave unveils a succinct mathematical principle within the latent space to describe images. It establishes a linear correlation between horizontal and vertical derivatives, expressed as:
> >
> > $\frac{\partial \mathbfit{\zeta}}{\partial x}=\mathbfit{\Lambda}\frac{\partial \mathbfit{\zeta}}{\partial y}$
> >
> > This formulation introduces shared wave speeds ($\mathbfit{\Lambda}$) across all images, revealing a mathematical invariance. Additionally, Hidden-Wave provides insights into how individual images differentiate from others through unique initial conditions while adhering to the same wave equations.
> >
> > ***Strength II: Describing the Entire Image.***
> > In contrast to the DCT, predominantly applied to 8x8 or 16x16 image blocks, Hidden-Wave is adept at describing the entire image.
> >
> > ***Strength III: Better Performance than DCT.***
> > Please refer to ***Conclusion II*** under **Weakness 1** in the first part of the rebuttal for more detailed information.
> >
> > ***Weakness: Not Straightforward.***
> > Despite its strengths, Hidden-Wave lacks the straightforwardness inherent in conventional compression methods like DCT. Its foundation on wave equations within a latent space introduces ambiguity about the meaning of each dimension. Additionally, Hidden-Wave requires an encoder and decoder to collaboratively learn, adding complexity to its implementation.
> >
> > ---
> >
> > **[Q3]: Is the selection of hyperparameters for training this method not very sensitive? How long does the training process take?**
> >
> > We only have two training hyperparameters (learning rate and weight decay). The selection of hyperparameters for training this method is not sensitive. All models are trained with the same hyperparameters.
> >
> > The training process takes varying amounts of time based on factors such as the number of channels (C), the number of special solutions (M), and the resolution of the feature map. The default setting (C=1024, M=4, feature map 64x64) takes 12 days with 8 V100 GPUs. Reducing the feature map size from 64x64 to 16x16 speeds up training to 7 days.

---

### Official Review · Reviewer_Zqsi · 2023-10-29

**Soundness:** 2 fair
**Presentation:** 3 good
**Contribution:** 2 fair
**Rating:** 5
**Confidence:** 3

**Summary:**

This paper studies a new finding about the mathematical properties of realistic images, i.e., that they can be recovered well from a set of specific solutions to one-way wave equations with the help of an encoder that generates initial conditions for the wave equations and a simple decoder that maps the wave equation solutions back to an image. The paper extends prior work of Chen et al., FINOLA, by interpreting their autoregressive process as the discretization of the one-way wave equation and by considering the sum of multiple such FINOLA solutions as an input to the decoder network. Numerical results illustrate that the reconstruction of the superposition of FINOLAs yields a significantly better PSNR than a single one, e.g., improving from 24.8 to 27.8.

**Strengths:**

The paper is well written and the continuous interpretation of FINOLA that implies a solution to a wave equation is very interesting. In particular, the fundamental type of question investigated here, i.e., if certain wave equations are a common mathematical principle that allows reconstructing any realistic image well is intriguing and has large potential if answered positively. In this sense, I found the paper very interesting to read.

**Weaknesses:**

Unfortunately, I also see several weaknesses in the presented work.
- Contribution: It is stated in section 2, that the prior work FINOLA is extended in two ways: a) generalizing FINOLA to a set of one-way wave equations and b) relaxing the local constraints. Yet, a) is merely an interpretation, and b) is done by summing over multiple FINOLA solutions. I consider this to be rather simple with the increase in PSNR being very natural due to the larger latent space of the resulting model.
- Insights on whether wave equations are really a common joint principle in natural images: There is no comparison to other autoencoders, no motivation on why the wave equation could be of particular importance to images, and no illustration/interpretation of the latent space encodings $q_i$ or the corresponding solutions $z_i$ of the autoregressive FINOLA features. In particular, a $256 \times 256 \times 3$ color image is represented, e.g. with a $8 \times 1024$ latent space variable and subsequently decoded (using the multi-path FINOLA) with a PSNR of 27.1. But how would approaches perform that use a latent space of similar dimension and do not exploit any autoregressive (or wave-equation-based) operation in their decoder? In order to believe that wave equations or FINOLA are a fundamental "underlying mathematical property shared by images", the proposed approach would have to perform significantly better than competing approaches. A truncated PCA, keeping the largest patch-wise DCT (or wavelet) coefficients of an image, and training a standard convolutional autoencoder would be natural very classical baselines. Currently, I am not convinced that multi-path FINOLA / wave equations are the fundamental ingredient that makes the autoencoder work exceptionally well.


After the rebuttal, I am increasing my score due to several very encouraging additional experiments. Yet, the new version of the paper is a major revision that should undergo a full new review process. In particular, technical details (like the architecture of the convolutional autoencoder) seem to be missing. In general,  in my opinion, to claim that wave equations are a mathematical property shared by all natural images more analysis is needed (including an interpretation of wave speeds, the diagonalizing basis, or experiments on what happens if the wave equation is solved differently from the special structure of the sum of FINOLAs). For claiming that a superposition of FINOLAs is one of the strongest latent compression techniques, more detailed comparisons (taking works on learned image compression into account) are needed.

**Questions:**

- As stated under 'Weaknesses' above, did you compare to any other encoding-decoding techniques that make the hypothesis, that wave equations are fundamental to all images more credible? The supplement compares to DCT and wavelet coding, but seemingly in terms of general aspects only, not in terms of reconstruction quality.
- The cited work of FINOLA by Chen et al. does not have a journal, conference or ArXiv ID. Is it published? Please complete the bibliography entry as this citation is of utmost importance for your work.

---

> ### Author Response · Authors · 2023-11-19
> **Authors' Rebuttal (part 1)**
>
> Thank you for dedicating your time and effort to provide feedback on our work. Below, we answer the questions that have been raised.
>
> ---
>
> **[Weakness 1]: Contribution: It is stated in section 2, that the prior work FINOLA is extended in two ways: a) generalizing FINOLA to a set of one-way wave equations and b) relaxing the local constraints. Yet, a) is merely an interpretation, and b) is done by summing over multiple FINOLA solutions. I consider this to be rather simple with the increase in PSNR being very natural due to the larger latent space of the resulting model.**
>
> Thank you for this comment. Let us clarify the contribution of Hidden-Wave over FINOLA as follows:
>
> ***I. Improved Mathematical Description:*** Hidden-Wave (one-way wave equations) enhances the mathematical description of images by generalizing FINOLA. Unlike FINOLA, which connects the partial derivatives of the feature map $\mathbfit{z}(x, y)$ along the $x$ and $y$ axes using the current feature values, Hidden-Wave directly links these derivatives through one-way wave equations:
>
> |Method|PDEs|
> |---|---|
> |FINOLA| $\frac{\partial \mathbfit{z}}{\partial x}=\mathbfit{A}\mathbfit{z}_n, \frac{\partial \mathbfit{z}}{\partial y}=\mathbfit{B}\mathbfit{z}_n$ |
> | **Hidden Wave (our)** | $\frac{\partial \mathbfit{z}}{\partial x}=\mathbfit{A}\mathbfit{B}^{-1}\frac{\partial \mathbfit{z}}{\partial y}$ |
>
> This generalization is *NOT* merely interpretative. We further demonstrate that a superior solution of wave equations can be *achieved* by summing multiple special solutions of FINOLA, validating that, for ***the same feature map size***, one-way wave equations offer a more accurate reconstruction than the two PDEs in FINOLA. This core insight underscores the paper's contribution:
>
> *"Mathematically, the feature map governed by one-way wave equations is more accurate than the one governed by the two PDEs in FINOLA."*
>
> ***II. Non-Trivial Summation over Multiple FINOLA:*** Summing over multiple FINOLA provides a simple yet elegant empirical validation of the generalized wave equations' optimality compared to the two PDEs in FINOLA. The feature map after summation stays within the solution space of wave equations through linearity, but departs from the solution space of FINOLA.
>
> ***III. Larger Latent Space:***
> This observation is invaluable! It sheds light on why multiple FINOLA paths effortlessly navigate the solution space of wave equations for a more optimal solution. Encoding more initial conditions simplifies the approach toward the optimal solution within the wave equation space.
>
> Moreover, multiple FINOLA paths provide a *parameter-efficient* means to adjust compression rate and reconstruction quality. In contrast to vanilla FINOLA, which achieves increased reconstruction quality by enlarging the number of channels and the sizes of matrices $\mathbfit{A}$ and $\mathbfit{B}$, multiple FINOLA achieves this by adding paths (or initial conditions) for the same number of wave equations and parameters in matrices $\mathbfit{A}$ and $\mathbfit{B}$. The table below compares Hidden-Wave with FINOLA across multiple latent sizes, ranging from 512 to 4096. Both methods achieve higher PSNR by increasing the latent size differently (by increasing paths in Hidden-Wave vs. increasing channels in FINOLA). Although Hidden-Wave is slightly behind FINOLA in terms of PSNR, it maintains a constant size for matrices $\mathbfit{A}$ and $\mathbfit{B}$, which is 256 times smaller than FINOLA at latent size 4096.
>
> |Method|Latent Size|DIM of $\mathbfit{A}$, $\mathbfit{B}$&darr;|PSNR&uarr;|
> |---|---|---|---|
> |FINOLA | 512 | 512x512| **22.0** |
> | **Hidden Wave (our)** | 512 (256x2) | **256x256 (x0.250)**| 21.8 |
> ||||
> |FINOLA| 1024 | 1024x1024| **23.4**|
> | **Hidden Wave (our)** | 1024 (256x4) | **256x256 (x0.063)** | 22.9 |
> ||||
> |FINOLA| 2048 | 2048x2048| **24.8**|
> | **Hidden Wave (our)** | 2048 (256x8) | **256x256 (x0.016)** | 24.4 |
> ||||
> |FINOLA| 4096 | 4096x4096| **25.9**|
> | **Hidden Wave (our)** | 4096 (256x16) | **256x256  (x0.004)** | 25.4 |

---

> > ### Author Response · Authors · 2023-11-19
> > **Authors' Rebuttal (part 2)**
> >
> > ***IV. Constant Complexity Decoding:***
> > Hidden-Wave handles a larger latent size elegantly without introducing additional parameters, due to the parameter sharing over the multiple FINOLA paths. This stands in contrast to vanilla FINOLA and other encoding-decoding techniques (e.g., DCT, DWT, autoencoder), which augment complexity to decode larger latent sizes. This feature allows Hidden-Wave to achieve comparable performance to FINOLA with significantly fewer parameters.
> >
> > Below is a comparison between FINOLA and Hidden-Wave under the same latent size. Hidden-Wave achieves similar (or slightly lower) PSNR with significantly fewer parameters in $\mathbfit{A}$ and $\mathbfit{B}$:
> >
> > |Method|Latent Size|DIM of $\mathbfit{A}$, $\mathbfit{B}$&darr;|PSNR&uarr;|
> > |---|---|---|---|
> > |FINOLA | 512 | 512x512| **22.0** |
> > | **Hidden Wave (our)** | 512 (256x2) | **256x256 (x0.25)**| 21.8 |
> > | **Hidden Wave (our)** | 512 (128x4) | **128x128 (x0.06)**| 21.5 |
> > ||||
> > |FINOLA| 1024 | 1024x1024| **23.4**|
> > | **Hidden Wave (our)** | 1024 (512x2) |  **512x512 (x0.25)** | 23.2 |
> > | **Hidden Wave (our)** | 1024 (256x4) | **256x256 (x0.06)** | 22.9 |
> > ||||
> > |FINOLA| 2048 | 2048x2048| **24.8**|
> > | **Hidden Wave (our)** | 2048 (1024x2) |  **1024x1024 (x0.25)** | **24.8** |
> > | **Hidden Wave (our)** | 2048 (512x4) | **512x512 (x0.06)** | 24.4 |
> > ||||
> > |FINOLA| 4096 | 4096x4096| 25.9|
> > | **Hidden Wave (our)** | 4096 (2048x2) |  **2048x2048 (x0.25)** | 25.9 |
> > | **Hidden Wave (our)** | 4096 (1024x4) | **1024x1024  (x0.06)** | **26.1** |
> >
> > ***V. Scaling Up Latent Size:***
> > The constant decoding complexity enables comparisons with other encoding-decoding techniques (DCT, DWT, autoencoder) with larger latent sizes, as shown later in **[Weakness 2 + Q1]**. This is particularly challenging for vanilla FINOLA.
> >
> > To illustrate, consider the case of '2048x8', where Hidden-Wave achieves the highest PSNR of 27.8. In contrast, the FINOLA counterpart would have 1x16384 channels, resulting in a staggering 16384x16384=268 million parameters in matrices $\mathbfit{A}$ and $\mathbfit{B}$. The computational cost would be 1 trillion FLOPs just for generating a feature map with a size of 64x64x16384. On the other hand, Hidden-Wave incurs only 4 million parameters (2048x2048) and 137 billion FLOPs for generating a feature map with a size of 64x64x2048.

---

> > > ### Author Response · Authors · 2023-11-19
> > > **Authors' Rebuttal (part 3)**
> > >
> > > **[Weakness 2 + Q1]: Comparing with other  encoding-decoding techniques  that use a latent space of similar dimension.**
> > >
> > > Thank you for this excellent suggestion. Below, we compare Hidden Wave with (a) DCT, (b) DWT/DTCWT, and (c) convolutional autoencoder for image reconstruction on the ImageNet validation set, with an image size of 256x256. Hidden Wave achieves a higher PSNR for the same or smaller latent size.
> > >
> > > ***Comparison with discrete cosine transform (DCT):*** The table below compares Hidden Wave with DCT. DCT is conducted per 8x8 image block, and the top-left $K$ coefficients (in zig-zag manner) are kept, while the rest are set to zero. We choose four $K$ values (1, 3, 6, 10) for comparison. Clearly, Hidden Wave achieves a higher PSNR with a small latent size. Notably, in the last row, *"mix"* refers to placing 8 initial conditions at different positions rather than overlapping at the center.
> > >
> > > |Method|Latent Size&darr;|PSNR&uarr;|
> > > |---|---|---|
> > > |DCT (top-left 1) | 3072 (32x32x3) | 20.6 |
> > > | **Hidden Wave (our)** | **2048 (1024x2)** | **24.8** |
> > > ||||
> > > |DCT (top-left 3)| 9216 (32x32x9) | 23.5|
> > > | **Hidden Wave (our)** | **8192 (1024x8)** | **27.1** |
> > > ||||
> > > |DCT (top-left 6)| 18432 (32x32x18) | 25.6|
> > > | **Hidden Wave (our)** | **16384 (2048x8)** | **27.8** |
> > > ||||
> > > |DCT (top-left 10)| 30720 (32x32x30) | 27.5|
> > > | **Hidden Wave (our)** | **16384 (2048x8 mix)** | **28.9** |
> > >
> > > ***Comparison with discrete wavelet transform (DWT) and dual-tree complex wavelet transform (DTCWT):*** We compare Hidden-Wave with DWT and DTCWT in the table below. Three scales are chosen for wavelet decomposition. The comparisons are organized into three groups: (a) using only the LL subband at the coarsest scale (scale 3), (b) using all subbands (LL, LH, HL, HH) at the coarsest level, and (c) using all subbands at the finer scale (scale 2). Hidden-Wave outperforms DWT and DTCWT in terms of PSNR for the first two groups, achieving a smaller latent size. In the last group, while Hidden-Wave's PSNR is lower than DTCWT, its latent size is significantly smaller (more than 10 times smaller).
> > >
> > > |Method|Latent Size&darr;|PSNR&uarr;|
> > > |---|---|---|
> > > |DWT (scale-3 LL) | 3888 | 21.5 |
> > > |DTCWT (scale-3 LL) | 12288 | 22.3 |
> > > | **Hidden Wave (our)** | **2048 (1024x2)** | **24.8** |
> > > ||||
> > > |DWT (scale-3 LL+LH+HL+HH)| 15552 | 24.3|
> > > |DTCWT (scale-3 LL+LH+HL+HH) | 49152 | 25.6 |
> > > | **Hidden Wave (our)** | **8192 (1024x8)** | **27.1** |
> > > ||||
> > > |DWT (scale-2 LL+LH+HL+HH)| 55953 | 28.7|
> > > |DTCWT (scale-2 LL+LH+HL+HH) | 196608 | **30.8** |
> > > | **Hidden Wave (our)** | **16384 (2048x8 mix)** | 28.9 |
> > >
> > > ***Comparison with convolutional Autoencoder (Conv-AE):*** The table below presents a comparative analysis between our method and convolutional autoencoder (Conv-AE) concerning image reconstruction, measured by PSNR. Both approaches share the same Mobile-Former encoder and have identical latent sizes (1024x2 or 1024x8). In our method, FINOLA is initially employed to generate a 64x64 feature map, followed by a convolutional decoder to reconstruct an image with a size of 256x256. On the other hand, Conv-AE employs a deeper decoder that utilizes convolution and upsampling from the latent vector to reconstruct an image. Hidden-Wave has significantly fewer parameters in the decoder. The results highlight the superior performance of FINOLA over Conv-AE, indicating that a single-layer FINOLA is more effective than a multi-layer convolution and upsampling approach.
> > >
> > > |Method|Latent Size&darr;|Params in Decoder&darr;|PSNR&uarr;|
> > > |---|---|---|---|
> > > | Conv-AE |1024x2| 35.9M | 24.6|
> > > | **Hidden Wave (our)** |1024x2| **16.6M**|**24.8** |
> > > ||||
> > > | Conv-AE | 1024x8 | 61.9M|26.0 |
> > > | **Hidden Wave (our)** | 1024x8 | **16.6M**|**27.1** |
> > >
> > > In summary, these new comparisons with three types of encoding-decoding techniques support our hypothesis that wave equations are fundamental to describe all images.
> > >
> > > ---
> > > **[Q2]: The cited work of FINOLA by Chen et al. does not have a journal, conference or ArXiv ID. Is it published? Please complete the bibliography entry as this citation is of utmost importance for your work.**
> > >
> > > FINOLA is an arxiv paper at the link below. We will complete the bibliography entry.
> > >
> > > *"Image as First-Order Norm+Linear Autoregression: Unveiling Mathematical Invariance, by Chen et al."*
> > > https://arxiv.org/abs/2305.16319.

---

> > > > ### Comment · Reviewer_Zqsi · 2023-11-22
> > > > **Interesting!**
> > > >
> > > > Thanks for the additional experiments! They are very encouraging! I will consider raising my score. Yet, there are quite some changes to be made to the paper to incorporate the new experiments into a more complete story (for which some literature on how much autoencoders can typically compress images could be cited as well). All in all, this seems to be a major revision for me, which should be reviewed again before publication. Yet, I find the general idea very interesting and encouraging and I like the overall work!
> > > >
> > > > As a side note: A non-FINOLA-based solution to the wave equation would have been a further interesting test. In terms of the final algorithm, it still feels like the sum of something existing/working (FINOLA) does a significant part of the job. Of course, this test is out of scope now (specifically now that I am posting this comment quite late - sorry for that).

---

> > > > > ### Author Response · Authors · 2023-11-22
> > > > > **Response to Reviewer Feedback and Planned Revisions**
> > > > >
> > > > > Thank you for your invaluable feedback and encouraging comments on our paper. Your time and effort in reviewing our work are highly appreciated.
> > > > >
> > > > > **Major Revision**
> > > > >
> > > > > We want to assure you that we are taking your feedback seriously. Currently, we are in the process of implementing a major revision to the paper. This includes incorporating the additional experiments you mentioned and integrating relevant literature on autoencoder-based image compression. The revised version will be submitted today, well before the deadline. We are committed to refining the paper to ensure a more complete and cohesive narrative, and we're pleased to hear that you find the general idea interesting and encouraging.
> > > > >
> > > > > **Non-FINOLA-based solution**
> > > > >
> > > > > Regarding your side note on a non-FINOLA-based solution to the wave equation, we appreciate your suggestion. We agree that it would have been a valuable addition to our study, and we plan to include it as a future perspective in the paper. The exploration of non-FINOLA-based solutions will be a focal point in our upcoming research.

---

> ### Author Response · Authors · 2023-11-23
> **The Revised Paper has been Submitted.**
>
> Dear Reviewer Zqsi,
>
> We are pleased to inform you that we have submitted the revised version of our paper as promised in our previous communication. In this revision, we have extensively reworked the experimental section, incorporating all experiments and discussions addressed during the rebuttal. Additionally, we have provided further clarification on the diagonalization of $\mathbfit{A}\mathbfit{B}^{-1}$ in Section 2.4. Furthermore, we have included visualizations and a comprehensive discussion on future work in the appendix.
>
> We greatly appreciate the invaluable feedback you provided, which has been instrumental in enhancing the quality of our work.
>
> Sincerely,
>
> -authors

---

### Official Review · Reviewer_kuax · 2023-10-30

**Soundness:** 3 good
**Presentation:** 3 good
**Contribution:** 2 fair
**Rating:** 6
**Confidence:** 4

**Summary:**

This article studies how to use wave equations in a hidden space of an image to recover it. The wave equations can be solved using first-order autoregressive generative model. The main contribution is to make two significant extensions of existing works to recover diverse images of high-resolution, one based on diagonalization of wave equations, the other based on using several first-order autoregressive models with shared weights.

**Strengths:**

The diagonalization idea is interesting as it simplifies the way to solve wavelet equations. As a whole, this model defines a powerful way to represent natural images with limited number of parameters.

**Weaknesses:**

A major issue is the lack of mathematical clarity in presenting your model. As a consequence, certain central idea such as diagonalization does not seem to be correct.

**Questions:**

-	In your model, A,B matrices defined in eq 2 are real-valued matrices and learnable. How do you guarantee that AB^-1 is diagonalizable such that the eigenvalues are all real-valued? (the Lambda diagonal matrix in eq 4 contains only real values along diagonal). In general, one could only assume that AB^-1 is diagonalizable such that V and Lambda are complex-valued (if AB^-1 is not symmetric).
-	It is not very clear in Fig 1 which part is trainable. Do you also train the encoder and decoder to recover input images ? If the encoder is not trained, how is it defined? Also what is your training loss?
-	Your decoder is linear, would that make the recovered images loss details (such as sharp edges) in images? This is not very clear from PNSR metrices. It would be better to add some visual examples.

---

> ### Author Response · Authors · 2023-11-19
> **Authors' Rebuttal (part 1)**
>
> Thank you for dedicating your time and effort to provide feedback on our work. Below, we answer the questions that have been raised.
>
> ---
>
> **[Q1]: In your model, A,B matrices defined in eq 2 are real-valued matrices and learnable. How do you guarantee that AB^-1 is diagonalizable such that the eigenvalues are all real-valued? (the Lambda diagonal matrix in eq 4 contains only real values along diagonal). In general, one could only assume that AB^-1 is diagonalizable such that V and Lambda are complex-valued (if AB^-1 is not symmetric).**
>
> Thank you for your valuable feedback. Allow us to provide further clarification regarding the diagonalization process. In equation 4, it's crucial to note that we do *NOT* impose any constraints on the eigenvalues, allowing them to be complex-valued in the diagonalization of $\mathbfit{A}\mathbfit{B}^{-1}$. In other words, the wave speed can take complex values. It's important to emphasize that $\mathbfit{\Lambda}$ and $\mathbfit{V}$ are *Not* explicitly trainable; instead, they are computed from the trainable matrices $\mathbfit{A}$ and $\mathbfit{B}$ through diagonalization post-training. We meticulously examined the eigenvalues in $\mathbfit{\Lambda}$ and eigenvectors in $\mathbfit{V}$ across multiple models after training, confirming their complex nature.
>
> In Section 2.4 of the paper, we also discuss an alternative approach wherein we constrain $\mathbfit{\Lambda}$ and $\mathbfit{V}$ to be real-valued by constraining matrices $\mathbfit{A}$ and $\mathbfit{B}$ as follows:
>
> $\mathbfit{A}=\mathbfit{P}\mathbfit{H}_x, \quad \mathbfit{B}=\mathbfit{P}\mathbfit{H}_y$
>
> where $\mathbfit{P}$ is the shared projection matrix for $\mathbfit{A}$ and $\mathbfit{B}$, and $\mathbfit{H}_x$ and $\mathbfit{H}_y$ are two real-valued diagonal matrices that are learnable. Consequently, the wave speed $\mathbfit{\Lambda} = \mathbfit{H}_x\mathbfit{H}_y^{-1}$ is also real-valued.
>
> However, it's noteworthy that enforcing real values for the wave speed leads to a performance drop in reconstruction quality. Table 2 (or the table below) compares the reconstruction PSNR between complex and real speeds, showing a slight decrease from 26.1 to 24.9.
>
> |Speed Value|\#Channels $C$|\#Solutions $M$|PSNR|
> |---|---|---|---|
> |Complex Value| 1024|4|**26.1**|
> |Real Value| 1024|4|24.9|
>
> ---
>
> **[Q2]: It is not very clear in Fig 1 which part is trainable. Do you also train the encoder and decoder to recover input images ? If the encoder is not trained, how is it defined? Also what is your training loss?**
>
> ***Trainable Components:*** The trainable elements in our architecture encompass both the encoder and decoder. The matrices $\mathbfit{A}$ and $\mathbfit{B}$ for the multi-path FINOLA are also trainable. Throughout training, the entire network is optimized end-to-end. It's essential to note that the wave speed $\mathbfit{\Lambda}$ and the coefficient matrices $\mathbfit{H}_x$ and $\mathbfit{H}_y$ are not explicitly trainable. Instead, they are computed post-training through the diagonalization process using the trainable matrices $\mathbfit{A}$ and $\mathbfit{B}$.
>
> ***Training of Encoder and Decoder*** Indeed, both the encoder and decoder undergo training to facilitate the reconstruction of input images.
>
> ***Training Loss:*** The training process employs mean square error over image pixels as the loss function, ensuring the end-to-end optimization of the entire network.

---

> > ### Author Response · Authors · 2023-11-19
> > **Authors' Rebuttal (part 2)**
> >
> > **[Q3]: Your decoder is linear, would that make the recovered images loss details (such as sharp edges) in images? This is not very clear from PNSR metrices. It would be better to add some visual examples.**
> >
> > Thank you for your insightful recommendation. Visual examples serve as compelling evidence, offering a more nuanced perspective beyond the limitations of PSNR. Please find below a link to three visualizations:
> >
> > Link to visualizations: https://tinyurl.com/yc5bb7dn
> >
> > ***(a) Comparison between Hidden-Wave ($M=8$) and FINOLA ($M=1$):***
> > The visual comparison vividly illustrates that Hidden-Wave, with a greater number of special solutions, achieves superior reconstruction quality by preserving finer visual details compared to FINOLA.
> >
> > ***(b) Reconstruction examples for varying number of wave equations ($C$) and special solutions ($M$):***
> > The visual examples provide a clear demonstration that increasing the number of special solutions consistently enhances image quality across different dimensions ($C=128$ to $C=2048$).
> >
> > ***(c) Reconstruction examples for complex-valued and real-valued wave speeds:***
> > This set of visualizations reveals that while complex-valued wave speed leads to superior image quality, the use of real-valued speed also results in reasonably good reconstructions.
> >
> > It's essential to note that despite the linearity of the one-way wave equations, the entire decoder is *NOT* purely linear. The decoder incorporates convolutional layers during the upsampling process from the feature map (resolution 64x64) to the reconstructed images at size 256x256. This non-linearity in the decoder is crucial for capturing and preserving intricate details, such as sharp edges, in the reconstructed images.

---

> > > ### Comment · Reviewer_kuax · 2023-11-22
> > >
> > > Thanks for your answers. I shall maintain my score for the following 2 reasons,
> > > - The notation in the article was not mathematically rigorous enough for me to understand the model. An improvement is needed. I still do not understand why xi(x,y) is in R^C (defined on top of page 4).
> > > - I do not know how to motivate the study of real-valued eigenvalues lambda_k in Table 2, as you are assuming that A B^-1 is diagonalizable in your model. It is not clear to me whether this assumption still holds when you consider real-valued eigenvalues.

---

> > ### Comment · Reviewer_Zqsi · 2023-11-22
> > **Short remark on diagonalization**
> >
> > In the discussion of [Q1], it is of course still not fully clear (or mathematically guaranteed) that $AB^{-1}$ is diagonalizable (even over $\mathbb{C}$), right? This just happens to be the case because the set of matrices that can not be diagonalized is so small/thin, that a non-diagonalizable matrix never occurs as a result of a learning process?

---

> > > ### Author Response · Authors · 2023-11-22
> > > **Diagonalizability and Disentanglement in Wave Equations (response to Reviewer Zqsi)**
> > >
> > > **[Second Round Question from Reviewer Zqsi]: In the discussion of [Q1], it is of course still not fully clear (or mathematically guaranteed) that $\mathbfit{A}\mathbfit{B}^{-1}$ is diagonalizable (even over $\mathbb{C}$), right? This just happens to be the case because the set of matrices that can not be diagonalized is so small/thin, that a non-diagonalizable matrix never occurs as a result of a learning process?**
> > >
> > > Indeed, your comment is accurate. The diagonalizability of $\mathbfit{A}\mathbfit{B}^{-1}$ is *NOT* guaranteed, even over complex values $\mathbb{C}$. In our experiments, however, this (diagonalizable) happens as a result of a learning process mainly because the set of matrices that can not be diagonalized is so small.
> > >
> > > This led us to elaborate on the connection between the ***diagonalizability*** of $\mathbfit{A}\mathbfit{B}^{-1}$ and the ***disentanglement*** of one-way wave equations.
> > >
> > > ***Nondiagonalizable $\mathbfit{A}\mathbfit{B}^{-1}$ &rarr; Entangled One-Way Wave Equations:***
> > > When $\mathbfit{A}\mathbfit{B}^{-1}$ is *not diagonalizable*, a linear partial differential equation (PDE) still holds:
> > >
> > > $\frac{\partial \mathbfit{z}}{\partial x}=\mathbfit{A}\mathbfit{B}^{-1}\frac{\partial \mathbfit{z}}{\partial y}$.
> > >
> > > This resembles a vectorized version of a one-way wave equation, with multiple dimensions linearly entangled.
> > >
> > > ***Diagonalizable $\mathbfit{A}\mathbfit{B}^{-1}$ &rarr; Disentangled One-Way Wave Equations:***
> > > In contrast, when $\mathbfit{A}\mathbfit{B}^{-1}$ is *diagonalizable*, the wave equations can be disentangled after diagonalization. Consequently, each dimension (after projection by the inverse of the eigenvector matrix) follows a one-dimensional wave equation:
> > >
> > > $\frac{\partial\zeta_k}{\partial x}=\lambda_k\frac{\partial \zeta_k}{\partial y}$
> > >
> > > Therefore, the diagonalizability of $\mathbfit{A}\mathbfit{B}^{-1}$ indicates if wave equations can be disentangled. In practice, our experiments suggest that non-diagonalizable matrices rarely occur.

---

> ### Author Response · Authors · 2023-11-22
> **Additional Mathematical Clarification (response to Reviewer kuax)**
>
> **[Second Round Q1]: why $\mathbfit{\zeta}(x,y) \in \mathbb{R}^C$ (defined on top of page 4)**
>
> Thank you for pointing out the notation discrepancy and expressing concerns about the mathematical rigor in our paper.
>
> The notation stating $\mathbfit{\zeta}(x, y) \in \mathbb{R}^C$ on top of page 4 was an oversight. We acknowledge this error and wish to clarify that $\mathbfit{\zeta}(x, y)$ is correctly defined as a complex-valued vector, denoted as $\mathbfit{\zeta}(x, y) \in \mathbb{C}^C$. This vector results from a linear projection of the real-valued feature map $\mathbfit{z}(x, y)$ using the inverse of the complex-valued eigenvalue matrix $\mathbfit{V}^{-1}$, as expressed by $\mathbfit{\zeta} = \mathbfit{V}^{-1}\mathbfit{z}$. Notably, $\mathbfit{V}^{-1}$ is itself a complex-valued matrix, as indicated by the relationship $\mathbfit{A}\mathbfit{B}^{-1} = \mathbfit{V}\mathbfit{\Lambda}\mathbfit{V}^{-1}$.
>
> We apologize for any confusion caused by this oversight and assure you that we are taking steps to enhance the clarity and accuracy of the mathematical expressions in our paper. If you have further questions or specific areas that require clarification, please feel free to let us know. Your feedback is crucial in improving the precision of our work.
>
> ---
>
> **[Second Round Q2]: I do not know how to motivate the study of real-valued eigenvalues lambda_k in Table 2, as you are assuming that A B^-1 is diagonalizable in your model. It is not clear to me whether this assumption still holds when you consider real-valued eigenvalues.**
>
> ***Motivation for Studying Real-Valued Eigenvalues:***
> The motivation behind our study of real-valued eigenvalues, specifically $\lambda_k$ in Table 2, stems from the question *"how the number type (real or complex) of wave speeds impacts reconstruction accuracy?"* By default, wave speeds are complex-valued, as previously discussed. To explore this, we conducted this ablation study, transitioning the number type from complex to real. The results of this investigation revealed a slight degradation in performance.
>
> ***Diagonalizable Assumption for Real-Valued Eigenvalues:***
> Addressing your concern about the assumption of diagonalizability when considering real-valued eigenvalues, we affirm that ***the diagonalizable assumption holds in our model***. We constrain the FINOLA matrices $\mathbfit{A}$ and $\mathbfit{B}$ as follows:
>
> $\mathbfit{A}=\mathbfit{P}\mathbfit{H}_x, \quad \mathbfit{B}=\mathbfit{P}\mathbfit{H}_y$
>
> where $\mathbfit{P}$ is a real-valued matrix, and $\mathbfit{H}_x$ and $\mathbfit{H}_y$ are real-valued diagonal matrices, all of which are learnable. This results in $\mathbfit{A}\mathbfit{B}^{-1} = \mathbfit{P}\mathbfit{H}_x\mathbfit{H}_y^{-1}\mathbfit{P}^{-1}$. Importantly, the eigenvalues $\mathbfit{\Lambda} = \mathbfit{H}_x\mathbfit{H}_y^{-1}$ and eigenvector $\mathbfit{P}$ are explicitly real-valued and learnable.
>
> We want to highlight that the only assumption in this context is that the projection matrix $\mathbfit{P}$ is invertible or full rank. Our experiments confirm that the learned $\mathbfit{P}$ matrix is indeed invertible.

---

### Official Review · Reviewer_nr9w · 2023-11-04

**Soundness:** 3 good
**Presentation:** 3 good
**Contribution:** 3 good
**Rating:** 8
**Confidence:** 2

**Summary:**

The paper proposes an extension of FINOLA to reconstruct images using one-way wave equations.

**Strengths:**

The theoretical intuition behind using multiple-path FINOLA is discussed thoroughly. They were also able to show that FINOLA is a special case of the method proposed in this paper. The quality seems to be consistently going up in the part of the graph showed to us.

**Weaknesses:**

In figure 4, it is not clear if there is plateau for the reconstruction quality in terms of M & C. The curves seem to be increasing linearly, and we do not see a slowdown

**Questions:**

See weakness

---

> ### Author Response · Authors · 2023-11-19
> **Authors' Rebuttal**
>
> Thank you for dedicating your time and effort to provide feedback on our work. Below, we answer the questions that have been raised.
>
> ---
>
> **[Weakness 1]: In figure 4, if there is plateau for the reconstruction quality in terms of M & C**
>
> Thanks for this insightful suggestion! We conducted additional experiments to explore the impact of increasing the number of special solutions (or the number of FINOLA paths $M$) to 16. We evaluated for three choices of the number of wave equations (or the number of channels $C$), specifically 128, 256, and 512. The PSNR results are presented in the last column of the table below. In comparison to the rate of change observed over smaller $M$ values (1→2, 2→4, 4→8), the PSNR increase from $M=8$ to $M=16$ slows down, indicating the onset of a plateau.
>
> ***Table: Reconstruction PSNR for different number of channels ($C$) and special solutions ($M$)***
> |Channels|**M=1**|**M=2**|**M=4**|**M=8**|**M=16**|
> |---|---|---|---|---|---|
> |**C=512**|  22.0|23.2|24.5|25.8|**26.3**|
> |**C=256**|  20.6|21.8|22.9|24.4|**25.5**|
> |**C=128**| 19.3|20.4|21.5|22.5| **23.1**|

---

> > ### Comment · Reviewer_nr9w · 2023-11-19
> >
> > I am so sorry about the previous submitted review. It looks an unfinished version of the review was submitted. Please check the updated review. But, your comment has already answered my question. Thank you.

---

### Author Response · Authors · 2023-11-21
**Reminder: Approaching End of Author-Reviewer Discussion Period**

Dear Reviewers,

We hope this message finds you well. We want to send a friendly reminder that the author-reviewer discussion period is nearing its close. Your input and perspective have been highly valuable throughout this process. If any questions or concerns about our rebuttal arise, we're here to listen and address them in a timely manner. Your insights are crucial in refining our manuscript further.

Understanding the demands on your time, we deeply appreciate your active participation in this rebuttal discussion. Your contributions are invaluable to us, and we genuinely thank you for your dedication.

Should you have any inquiries or thoughts, please don't hesitate to reach out.

Best regards,

-authors

---

### Author Response · Authors · 2023-11-23
**Submission of Revised Paper with Major Revisions**

Dear Reviewers,

We are delighted to inform you the submission of the revised version of our paper, which incorporates the valuable feedback and suggestions provided by each of you. The major revisions implemented are outlined below:

1. Extensive reworking of the experimental section (Section 3), integrating all experiments and discussions addressed during the rebuttal. [***Reviewer Zqsi and Y4BM***]
2. Addition of visualizations for reconstructed exemplars in Appendix C. [***Reviewer kuax***]
3. Inclusion of a discussion on future work in Appendix F. [***Reviewer Y4BM***]
4. Clarification on the diagonalization of $\mathbfit{A}\mathbfit{B}^{-1}$ in Section 2.4. [***Reviewer Zqsi and kuax***]
5. Presentation of results for $M=16$ in Appendix D to investigate the slowdown in PSNR increase over $M$. [***Reviewer nr9w***]

We express our sincere gratitude for the invaluable feedback you provided.

Sincerely,

-authors

---

### Meta-Review · Area_Chair_FnEX · 2023-12-06

**Metareview:**

The paper studies whether realistic images can be recovered well from a set of specific solutions to one-way wave equations. The paper builds on prior work by Chen et al., FINOLA.
The paper received mixed reviews. While the question investigated is very interesting, the reviewers raised the following concerns:

* Reviewer kuax (3) finds that a main issue is the lack of mathematical clarity, and after discussion still finds the paper lacking in rigor.

* Reviewer Zqsi (3) initially saw two issues that could be partly cleared up, however after rebuttal and reviewer discussion, more detailed comparisons are required for the claim that FINOLAs is one of the strongest latent compression techniques, and even after the rebuttal and discussion, important point about the paper are still unclear.

* Reviewer Y4BM (6) finds the understanding of FINOLA, a general wave representation of an image, developed interesting, but notes that the experimental design has issues, it doesn't lead meaningful insights or conclusions, and while the phenomenon is interesting, it is unclear how research or applications can be build on the paper.

Given that there are still issues regarding the clarity, rigor, and support for certain statements, the paper is not yet ready for publication.

**Justification For Why Not Higher Score:**

Issues with clarity, rigor, and contribution relative to prior work.

**Justification For Why Not Lower Score:**

N/A.

---

### Decision · Program_Chairs · 2024-01-16

Reject